# Simulating electron transfer on noisy quantum computers

**Marvin Gajewski** [1,2], **Alejandro D. Somoza** [1,2] ✉, **Gary Schmiedinghoff** [3], **Pascal Stadler** [4], **Michael Marthaler**[4] **& Birger Horstmann** [1,2,5] ✉

While simple spin-boson models have been realized on quantum hardware, simulating extended electronic networks with local vibrational environments remains a fundamental challenge in the presence of non-equilibrium, long-lived electronic-vibrational (vibronic) coherence. We present a framework for the digital-analog simulation of open quantum systems governed by Hamiltonians with linear-vibronic coupling (LVC) and structured vibrational environments. Our approach exploits the intrinsic dissipation of qubits in near-term quantum hardware as a resource to emulate vibrational relaxation, combined with a model-specific error mitigation scheme to filter out noise sources incompatible with the target open system. We validate our strategy by resolving the vibronic transfer spectra of a one-dimensional donor-acceptor chain on IBM superconducting processors, reproducing non-Markovian dynamics and scaling the chain length up to 10 electronic sites, an unprecedented scale for chemical dynamics on quantum computers. Our model of vibronic electron transfer offers a portable, application-oriented benchmark for simulating long-lived entangled states on NISQ computers.

The transformation of the energy and mobility sectors demands the development of novel and powerful simulation tools such as quantum computers, supporting the design of new materials operating across a wide range of environmental conditions. Kinetic theories at thermal equilibrium fail to accurately capture transfer rates in relevant scenarios, such as the inverted region in the Marcus theory of electron transfer[1,2]. This regime is for example relevant for batteries with large overpotentials, indicative of high reorganization energies and complex solvation shells[3,4] that are linked to dissipative processes. A proper understanding of heat loss at the microscopic scale is crucial in the engineering of next-generation devices for energy storage and production.

Nonequilibrium processes on the picosecond scale ($10^{-12}$ s) are coming into focus thanks to the increasing time resolution of spectroscopic techniques[5,6]. Nonequilibrium quantum effects are a promising route to increase the efficiency of energy materials thanks to their unusual thermodynamic properties[7], such as long-lived electronic-vibrational (vibronic) coherence[8,9] or the slow cooling of hot-carriers[10,11]. These processes are found to strongly affect the efficiency and directionality of electron transfer (ET) between reactant molecules and electrode surfaces[12–14], as well as the ultrafast charge separation in organic photovoltaics[15] or photosynthetic systems[16], and are also believed to strongly influence the transport of charges in battery electrodes[17,18] and perovskites[19].

Quantum computers bring the potential to simulate the nonequilibrium dynamics of materials in operando conditions to reveal new insights into the underlying mechanisms that govern their performance[20]. As such models are described by open quantum systems, noise processes in the quantum hardware that are compatible with the target open quantum system can be exploited[21–25], while unwanted sources of noise are mitigated insofar as possible[26,27]. There is a growing interest in using dissipation and structured noise as a resource in quantum computing, from the acceleration of variational optimization workflows[28] to the engineering of initial states[29–32]. In

[1]Institute of Engineering Thermodynamics, German Aerospace Center (DLR), Ulm, Germany. [2]Helmholtz Institute Ulm, Ulm, Germany. [3]Institute of Software Technology, German Aerospace Center (DLR), Köln, Germany. [4]HQS Quantum Simulations GmbH, Karlsruhe, Germany. [5]Department of Physics, Ulm University, Ulm, Germany. ✉e-mail: alejandro.somoza@dlr.de; birger.horstmann@dlr.de

particular, novel quantum platforms that naturally realize quantized oscillators via bosonic elements in the hardware have the potential to become powerful toolboxes for the simulation of open quantum systems[33–36], avoiding resource-intensive boson-to-qubit encodings[23]. On trapped ions, vibronic electron transfer with tunable dissipation has been demonstrated for dimer systems[37,38], as well as the engineering of structured environments for spin-boson models[39–41]. Although quantum algorithms for the simulation of extended networks like the Holstein model have been put forth[42–44], experimental demonstrations with large problem sizes are still notoriously challenging.

We focus on the non-equilibrium dynamics of vibronic networks, where ultrafast ET ($\lesssim 100$ fs) can be observed in the non-Markovian dynamics, whose duration is determined by the finite lifetime of vibrational motion[45]. While ultrafast ET can happen via purely electronic resonance between donor and acceptor states[46], it can remarkably also occur via vibronic resonance mediated by the coherent interaction of electrons and vibrations like C=C stretch modes in organic photovoltaics[47], whose frequencies are one order of magnitude above room temperature. Such effects are supported by pump-probe experiments showing clear evidence of long-lived ($\lesssim 0.5$ ps) vibronic coherence in the charge separation of electronic excitations[15,48–50].

Our approach uses the pseudomode formalism to simulate the dynamics of the open vibronic system[51–57], as depicted in Fig. 1a. Using auxiliary harmonic oscillators (pseudomodes) that are locally damped by their respective Markovian baths, one is able to engineer arbitrary spectral densities to simulate the non-Markovian dynamics of the reduced system of interest (the electronic network). One can substantially lower the memory requirements in numerical simulations when combining pseudomodes with tensor networks techniques[5,58,59], because local damping naturally destroys correlations between oscillators. This logic can be extended to quantum algorithms that exploit local dissipation as a quantum resource. By using damped oscillators in the simulation of open quantum systems on NISQ computers, entanglement requirements may be lowered and adapted to the lifetime of system-environment correlations[60]. While classical methods such as HEOM or T-TEDOPA are capable of simulating small systems with highly structured environments[61], they face severe limitations when scaling to larger networks, illustrating that the primary challenge in simulating open quantum systems lies in the scaling of the system size, showing the potential utility for quantum algorithms like the one presented here.

When simulating the Trotterized dynamics of the closed model on the quantum computer with a time step $\Delta t$ as shown in Fig. 1, the intrinsic noise of the quantum processor imprints an effective damping rate $\Gamma_{QC}$, which is determined by the $T_1$, $T_2$ times of the involved qubits (see Eq. (18)). Rather than targeting a specific damping rate, this work focuses on maximizing simulation time to identify the longest vibrational relaxation lifetime $\Gamma_{QC}$ achievable on the quantum

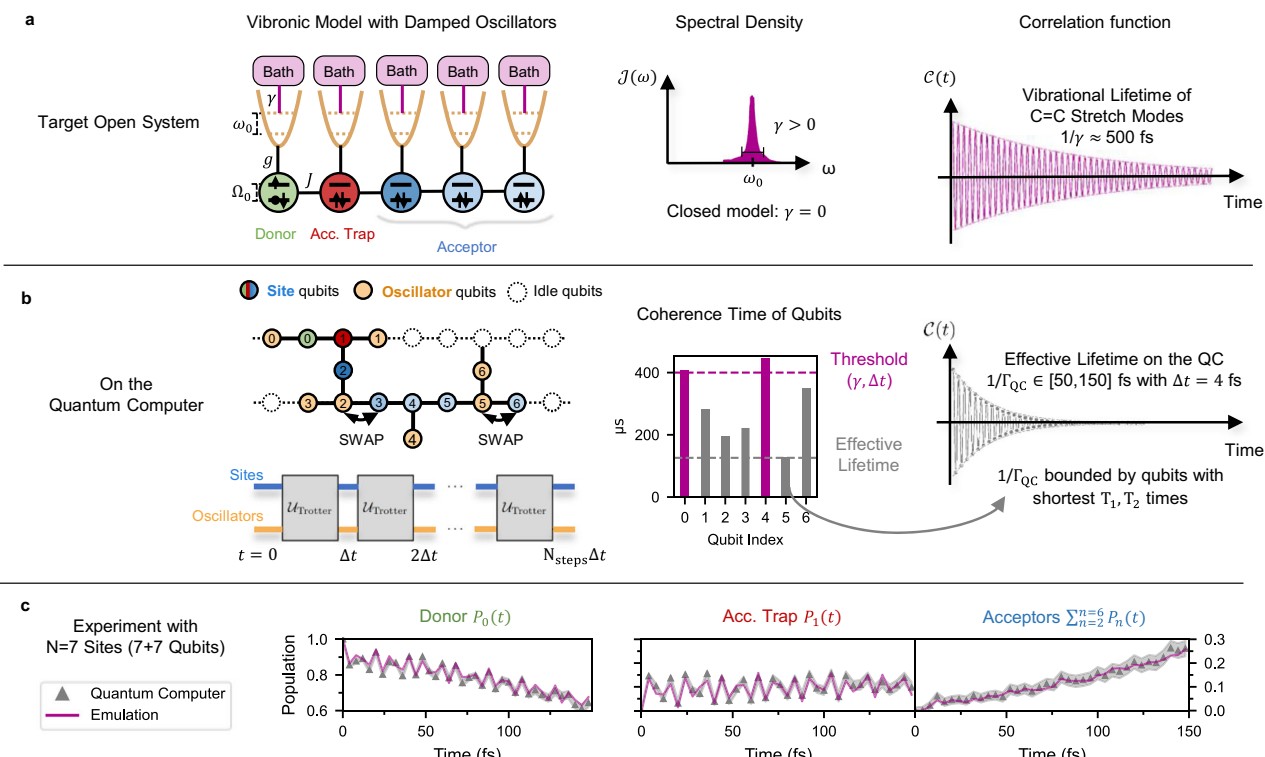

**Fig. 1 | Algorithm overview. a** The left panel shows a schematic of the one-dimensional LVC Hamiltonian with nearest-neighbors electronic interactions of coupling strength $J$, where each site of energy $\Omega_n$ is locally coupled to a single quantum harmonic oscillator of frequency $\omega_0$ with vibronic coupling strength $g$. These oscillators may be damped by independent vibrational environments with a rate $\gamma$ that are characterised by the width of the peak of the same spectral density $\mathcal{J}(\omega)$ (Eq. (5)), shown in the center panel. The lifetime of vibrational excitations is given by $1/\gamma$ and determines the duration of system-environment correlations (non-Markovian dynamics), which can be observed in the right panel with the bath correlation function $\mathcal{C}(t)$ (Eq. (13)). **b** On the quantum computer, each electronic site is mapped to a single qubit while oscillators are mapped to qubits using a boson-to-spin encoding (left panel). Trotterized circuits corresponding to the closed model ($\gamma = 0$) simulate the dynamics of the open model ($\gamma > 0$) with a time step $\Delta t$ due to the intrinsic damping of oscillator qubits. The center panel illustrates a distribution of coherence times of qubits involved in the circuit, together with the threshold required to simulate weakly damped C=C stretch modes. Note that the threshold depends on the choice of $\Delta t$ and the circuit execution time on the hardware (Eq. (18)). We found that the shortest $T_1$ or $T_2$ time determines the effective damping rate $\Gamma_{QC}$, that lies for the simulations presented here between $(50\,\text{fs})^{-1}$ and $(150\,\text{fs})^{-1}$ (right panel). **c** Experimental results (gray markers) for vibronic electron transfer in a chain of $N = 7$ sites (14 qubits) are compared to an emulation of the open quantum system (magenta), where all oscillator qubits experience the same damping rate $\gamma = (100\,\text{fs})^{-1}$ (see Fig. 5d, e).

processor. The effective damping rate $\Gamma_{QC}$ can be tuned by adjusting the Trotter step $\Delta t$: larger steps reduce dissipation imprinted on the resulting dynamics, while smaller steps increase it, as demonstrated previously[23]. However, this tuning is constrained by two key limits. First, $\Delta t$ must be sufficiently small to fully resolve the spectral content of the Hamiltonian, determined by its largest possible frequency. Second, excessively small $\Delta t$ leads to deep quantum circuits, increasing the number of time steps required to reach a given evolution time. In general, target models with stronger damping rates ($\gamma \gg \Gamma_{QC}$) can also be implemented either via delay instructions on environment qubits[32] or alternatively, by coupling the latter to an ancillary qubit with periodic reset operations[62]. However, the simulation of target models with longer vibrational lifetimes ($\gamma \ll \Gamma_{QC}$) requires hardware with larger qubit coherence times beyond what is available today. For a faithful simulation only qubits that are used to encode oscillators should be subject to dissipation, while noise on other qubits requires mitigation.

In this work, we present a protocol to simulate non-equilibrium dynamics of vibronic networks, and validate the approach by simulating a microscopic model of electron transfer on the superconducting quantum computer IBM_AACHEN. The model consists of an electronic chain of donor and acceptor sites, each of which are coupled to a local vibrational mode. Our experimental results reproduce ET dynamics on up to 10 electronic sites (20 qubits) and lead to an effective lifetime $1/\Gamma_{QC}$ that lies between 50 and 150 fs (Fig. 1b), approaching the long-lived motion of intramolecular vibrational modes in organic molecules ($1/\gamma \gtrsim 500$ fs), a fundamental challenge in classical simulations of open quantum systems where low damping rates render perturbative methods highly inaccurate[5].

## Results

### Electron transfer model

We consider a microscopic model of electron transfer from a single donor to $N-1$ acceptor sites, coupled with nearest-neighbor electronic interactions (see Fig. 1a). The excitation initially sits at the donor site ($n=0$), and the first acceptor site ($n=1$) acts as an energetic trap that competes with successful charge separation. In the thermodynamic limit ($N \to \infty$), this constitutes the archetypal problem in the theory of electron transfer where a single donor is coupled to a continuous band of acceptor states[63].

The Hamiltonian parameters are chosen to model the ultrafast charge separation in a prototypical donor-acceptor blend from organic photovoltaics, consisting of a donor polymer like P3HT (Poly(3-hexylthiophene)) and functionalized fullerenes like PCBM ([6,6]-phenyl-C61-butyric acid methyl ester) as acceptor molecules[47]. As represented by colored circles in Fig. 1a, each molecule is treated as a two-level system that corresponds to its highest occupied (HOMO) and lowest unoccupied (LUMO) molecular orbitals. The coupling between LUMO levels leads to electronic hopping between neighboring sites.

The initial state is given by a single electron-hole pair at the donor site (shown in green in Fig. 2a), after which the electron may transfer to the acceptor molecules, while the hole is assumed to be frozen at the donor because of its lower mobility. This allows us to focus on the dynamics of the electron in the Coulomb potential that is induced by the hole, reflecting the bound nature of electron-hole pairs (excitons) in organic molecules. Charge separation is thought to take place faster than the creation of excitons via the absorption of photons in the donor. It is therefore common to assume that the dynamics remain in the single-excitation manifold at all times. The Hamiltonian incorporates both electronic and vibrational degrees of freedom, and is given by

$$\widehat{H} = \widehat{H}_{el} + \widehat{H}_{vib} + \widehat{H}_{el-vib}. \tag{1}$$

The electronic part of the Hamiltonian is modeled as a tight-binding chain $\widehat{H}_{el}$

$$\widehat{H}_{el} = \sum_{n=0}^{N-1} \Omega_n \widehat{a}_n^\dagger \widehat{a}_n + \sum_{n=0}^{N-2} J\left(\widehat{a}_n^\dagger \widehat{a}_{n+1} + \widehat{a}_{n+1}^\dagger \widehat{a}_n\right), \tag{2}$$

where $J = 500$ cm$^{-1}$ is the coupling strength between sites, $\Omega_0$ is the energy of the LUMO level of the single donor site, $\Omega_n = -V/n$ ($n \geq 1$) is the LUMO energy of the $n$-th acceptor site, $V = 2420$ cm$^{-1}$ is the Coulomb binding energy ($V = |\Omega_1 - \Omega_n|$ for $n \to \infty$) and $\widehat{a}_n^\dagger$ ($\widehat{a}_n$) are the creation (annihilation) operators for an electron on site $n$. The local energy levels $\Omega_n$ are visualized in Fig. 2a, where the first acceptor site ($n=1$) clearly constitutes an energetic trap. Note that despite the assumption of a single-excitation, the occupied Hilbert space of the vibronic Hamiltonian in Eq. (1) generally scales exponentially with the system size $N$ because of the vibronic terms.

The vibrational part of the Hamiltonian $\widehat{H}_{vib}$ couples each electronic site $n$ locally to a set of $M_n$ quantum harmonic oscillators

$$\widehat{H}_{vib} = \sum_{n=0}^{N-1} \sum_{m=0}^{M_n-1} \omega_{n,m} \widehat{b}_{n,m}^\dagger \widehat{b}_{n,m}, \tag{3}$$

where $\widehat{b}_{n,m}^\dagger$ and $\widehat{b}_{n,m}$ are the creation and annihilation operators for the $m$-th vibrational oscillator at site $n$, and frequency $\omega_{n,m}$. The linear vibronic coupling (LVC) term $\widehat{H}_{el-vib}$ describes the interaction between the electronic states and oscillators

$$\widehat{H}_{el-vib} = \sum_{n=0}^{N-1} \sum_{m=0}^{M_n-1} g_{n,m} \left(\widehat{a}_n^\dagger \widehat{a}_n \otimes (\widehat{b}_{n,m}^\dagger + \widehat{b}_{n,m})\right), \tag{4}$$

where $g_{n,m}$ is the vibronic coupling strength between the $n$-th site and the $m$-th vibrational oscillator.

The interaction between site $n$ and its local vibrational environment is captured in the spectral density function

$$\mathcal{J}_n(\omega) = \sum_{m=0}^{M_n-1} g_{n,m}^2 \delta(\omega - \omega_{n,m}), \tag{5}$$

that becomes a continuous function in the thermodynamic limit $M_n \to \infty$ of the bath. We focus on modeling high-frequency, dispersionless, intramolecular vibrations like C=C stretch modes that are intrinsic to organic molecules. Therefore, we assume that every site is coupled to identical vibrational environments that are independent from each other

$$\mathcal{J}_n(\omega) \equiv \mathcal{J}(\omega) \text{ for all } n \in [0, N-1], \tag{6}$$

and are characterized by a single Lorentzian peak around $\omega_{n,0} \equiv \omega_0$ with $\omega_0 = 1500$ cm$^{-1}$. Therefore, identifying $M_n \equiv M$, $\widehat{b}_{n,0} \equiv \widehat{b}_n$ and $g_{n,0} \equiv g_0$, we model $\mathcal{J}(\omega)$ with a single pseudomode (see Methods), i.e. a single damped oscillator ($M=1$) of frequency $\omega_0$ and vibronic coupling strength $g_0 = \omega_0 \sqrt{s}$, where $s = 0.05$ is the dimensionless Huang-Rhys factor. The damping rate $\gamma$ is determined by the width of the Lorentzian peak in the spectral density $\mathcal{J}(\omega)$ (see Fig. 1a).

The pseudomode formalism leads to the following master equation for the density matrix

$$\frac{d\widehat{\rho}}{dt} = -\frac{i}{\hbar}[\widehat{H}, \widehat{\rho}] + \sum_{n=0}^{N-1} \mathcal{L}_{vib}^{(n)}(\widehat{\rho}), \tag{7}$$

where $\widehat{H}$ is the full vibronic Hamiltonian in Eq. (1) and $\mathcal{L}_{vib}^{(n)}(\widehat{\rho})$ are the Lindblad superoperators describing the local damping of each oscillator. Although Eq. (7) takes the form of a Lindblad master equation, the pseudomode approach in principle allows to reproduce arbitrary

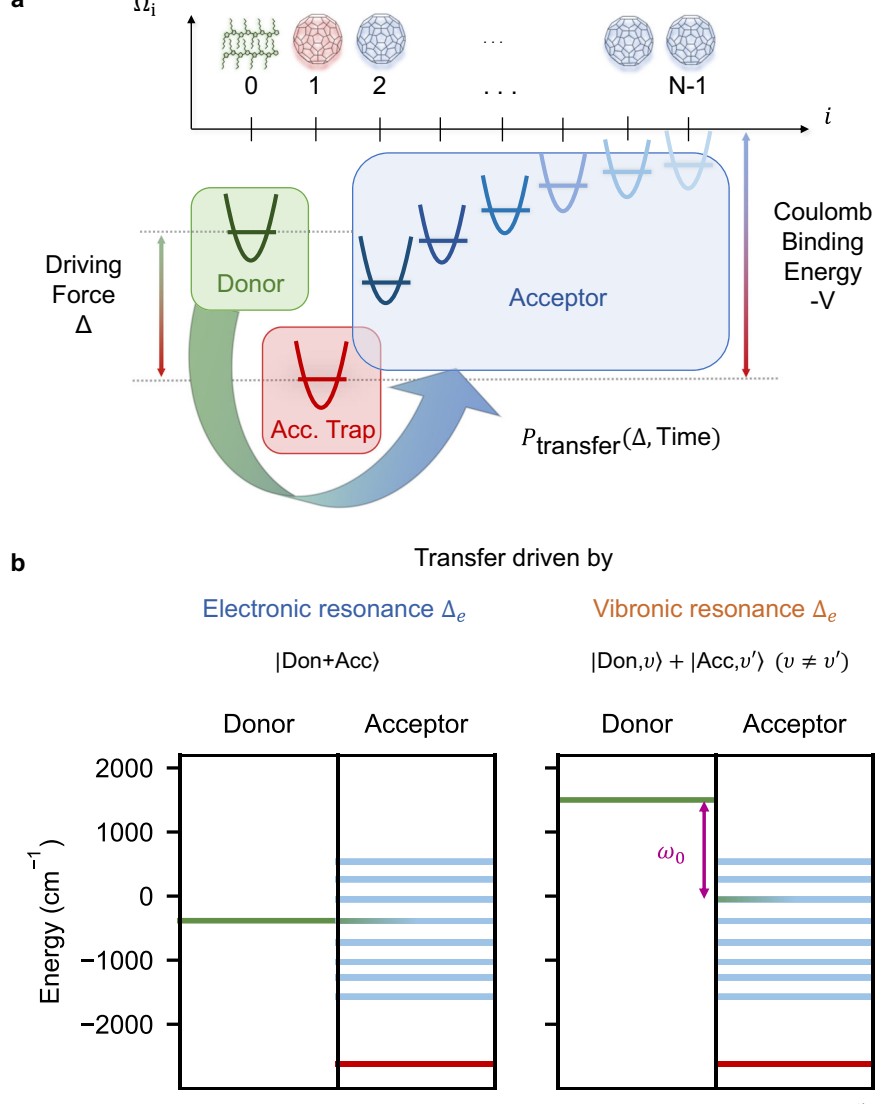

**Fig. 2 | Microscopic model of the donor-acceptor chain and mechanisms of electronic and vibronic ET. a** Schematic of the one-dimensional donor-acceptor interface with vibronic coupling to high-frequency, intramolecular C=C stretch modes. We vary the driving force $\Delta$ of the donor site ($n = 0$) to study the probability of electron transfer $P_{\text{transfer}}(\Delta, \text{Time})$ to the acceptor sites ($n > 1$). **b** Donor energy and eigenenergies of the acceptor subsystem $\widehat{H}_{\text{el}}^{(\text{Acc.})}$ (see Eq. (8)) for typical electronic and vibronic resonances. Both mechanisms induce ultrafast ET by creating superpositions of states with mixed donor/acceptor character. For the vibronic case (right-hand side), the superposition is created by coherent electron-oscillator (vibronic) coupling.

non-Markovian dynamics in the the electronic degrees of freedom ($\widehat{\rho}_{\text{el}} = \text{Tr}_{\text{vib}}\widehat{\rho}$), where the pseudomodes are traced out[51–53].

We probe two different mechanisms for ET: depending on the driving force $\Delta = \Omega_0 - \Omega_1$, ET occurs via electronic or vibronic resonances, as depicted in Fig. 2b. For purely electronic resonances, the donor energy $\Omega_0$ is resonant with an eigenenergy $\Delta_e$ of the acceptor subsystem

$$\widehat{H}_{\text{el}}^{(\text{Acc.})} = \sum_{n=1}^{N-1} \Omega_n \,\widehat{a}_n^\dagger \widehat{a}_n + \sum_{n=1}^{N-2} J\left(\widehat{a}_n^\dagger \widehat{a}_{n+1} + \widehat{a}_{n+1}^\dagger \widehat{a}_n\right), \qquad (8)$$

defined in the same manner as $\widehat{H}_{\text{el}}$, but restricted in the sums to $n > 0$ to exclude the donor site at $n = 0$. Vibronic coupling is not required for electronic resonances to occur. Vibronic resonances require vibronic coupling and occur for driving forces $\Delta_v \approx \Delta_e + \omega_0$, when the energy difference between the donor and one of the acceptor eigenstates of $\widehat{H}_{\text{el}}^{(\text{Acc.})}$ is matched by the energy $\omega_0$ of a vibrational excitation. The mechanism of vibronic ET is thus driven by a non-separable

(entangled) superposition of electronic and vibrational excitations (right panel in Fig. 2b). Further details are explained in Supplementary Notes.

**Quantum simulations, emulations and classical simulations**

This work aims at quantum simulations of the presented ET-Model with a single pseudomode per electronic site, for which $N_b = 2$ energy levels per oscillator are sufficient, as validated in the Supplementary Notes. We describe a more general qubit encoding, suitable for generic vibronic Hamiltonians with various oscillators and multiple energy levels, in the Supplementary Methods. We show the scalability of the approach in Discussion.

For a model with $N$ sites, $N$ qubits are used to simulate the electronic subspace and $N$ further qubits for the damped oscillators (pseudomodes). We perform *quantum simulations* of this model as depicted in Fig. 1a by using quantum circuits corresponding to the full Hamiltonian in Eq. (1), employing Trotterized time evolution on the quantum computer with $10^4$ shots per time step. Our model-specific

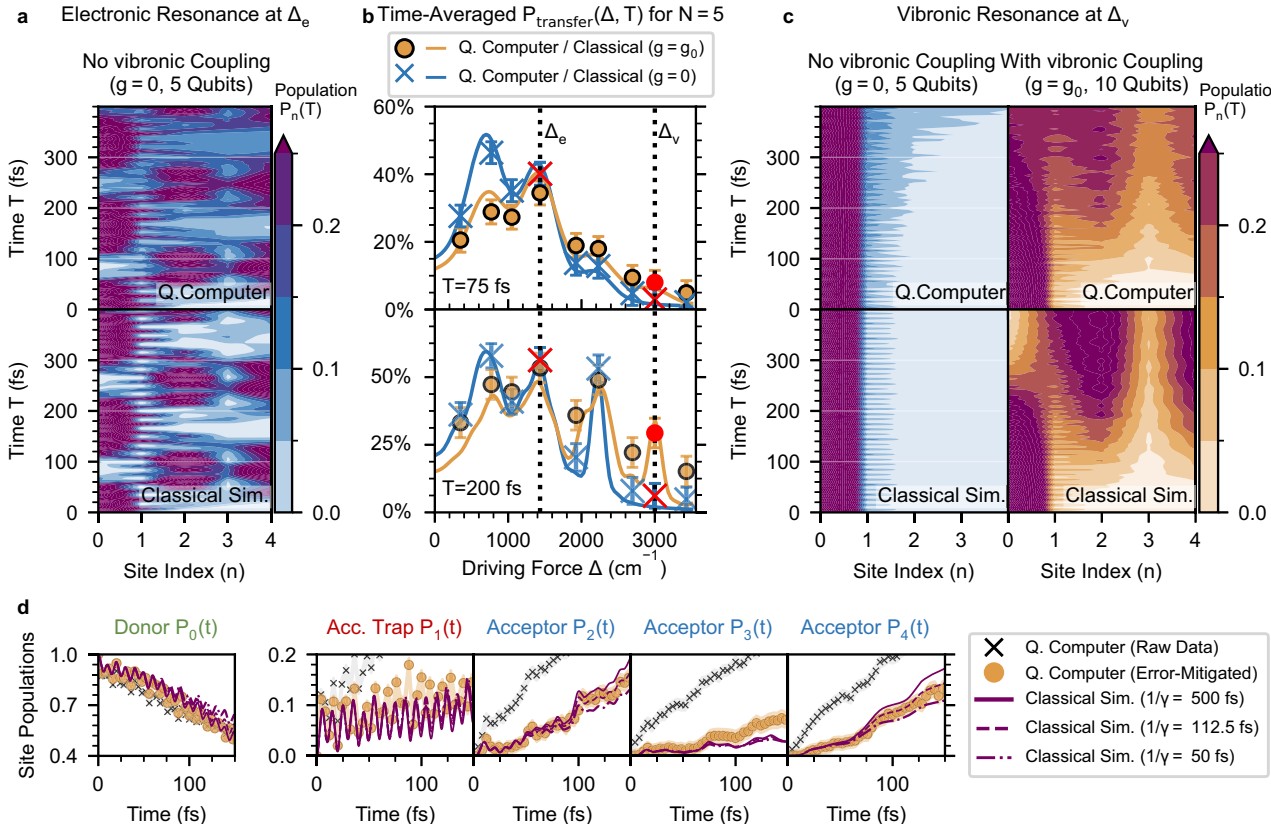

**Fig. 3 | Probing different charge transfer mechanisms for $N = 5$ sites on IBM_AA-CHEN. a** The top panel shows a quantum simulation of the population dynamics $P_n(t)$ for an electronic resonance at $\Delta_e = 1435$ cm$^{-1}$ without vibronic coupling ($g = 0$) on 5 qubits. The bottom panel shows a classical simulation ($\Delta t = 0.5$ fs) for reference. To enhance visibility for the population dynamics over few sites and many time steps, the contour plot shows lines of equal population, using a linear interpolation of the position coordinate. **b** Transfer probability $P_{transfer}(\Delta, T)$ for $T = 75$ fs (top panel) and $T = 200$ fs (bottom panel) for the cases with ($g = g_0$ in orange, 10 qubits) and without ($g = 0$ in blue, 5 qubits) vibronic coupling. The points show expectation values obtained from the experiment, with error bars accounting for shot noise. The

curves display classical simulations with a damping rate of $\gamma = (500$ fs$)^{-1}$. The simulations shown in **a** and **c** are marked in red. **c** The top panels shows a quantum simulation of the population dynamics $P_n(t)$ at the vibronic resonance ($\Delta_v = 3010$ cm$^{-1}$) without ($g = 0$, blue) and with ($g = g_0$, orange) vibronic coupling. The bottom panels show classical simulations ($\Delta t = 0.5$ fs, $N_b = 5$) of the target model with a damping rate of $(112.5$ fs$)^{-1}$ as reference. **d**. Population dynamics $P_n(t)$ of the run with $\Delta_v$ and $g = g_0$ (also shown in **c**) with and without error mitigation, and three classical simulations with increasing damping rates of $(500$ fs$)^{-1}$ (solid lines), $(112.5$ fs$)^{-1}$ (dashed lines) and $(50$ fs$)^{-1}$ (dashed-dotted lines). Each time point corresponds to a unique quantum circuit sampled with $10^4$ shots.

error-mitigation scheme, which is based on discarding shots from the raw measurement results, filters out sources of noise like depolarization that are incompatible with the pseudomode formalism while retaining damping in the hardware qubits (see Methods). The resulting ET dynamics on the quantum computer exhibits an effective damping rate $\Gamma_{QC} > 0$ that is strongly influenced by the qubits with the worst properties in the circuit, such as short coherence time and large gate errors (see Fig. 1b). Note that we attempt to reach the longest possible lifetime by choosing the qubits with the best properties, instead of targeting a specific damping rate.

To validate the performance of our quantum simulations, we compare against two different types of numerical simulations. We refer to numerical simulations of the quantum circuits ($\Delta t = 4$ fs, $N_b = 2$) as *emulations*. They involve no noise besides amplitude damping on qubits encoding oscillators and are obtained with the QuEST toolkit[64]. Simulations of the target model using a smaller time step $\Delta t = 0.5$ fs and $N_b = 5$ levels per oscillators are referred to as *classical simulations* and obtained using the Dissipation-Assisted Matrix Product Factorization (DAMPF) method, which efficiently applies the pseudomode approach in tensor networks[53,59]. Comparing our quantum simulations against emulations allows us to assess the effects of noise in the quantum computer, while classical simulations can be considered an exact benchmark with negligible Trotter

error and negligible oscillator truncation error (see Supplementary Notes).

## Probing electronic and vibronic resonances for $N = 5$ sites

We evolve an initial excitation in the donor site and aim for a qualitative reproduction of the time-averaged probability of electron transfer $P_{transfer}(\Delta, T)$,

$$P_{transfer}(\Delta, T) = \frac{1}{T} \sum_{n=2}^{N} \int_0^T P_n(t)\, d t, \qquad (9)$$

as a function of the driving force $\Delta$ and integration time $T$, where $P_n(t)$ is the population of site $n$. The transfer probability describes the probability of an electron moving from the donor to the acceptor without getting stuck in the trap, see Fig. 2a. We expect two types of peaks in $P_{transfer}(\Delta, T)$, associated with the electronic and vibronic mechanism of ET illustrated in Fig. 2b.

In Fig. 3a we show error-mitigated experimental results for the site population dynamics $P_n(t)$ on IBM_AACHEN at an electronic resonance with driving force $\Delta_e = 1435$ cm$^{-1}$ for the case without vibronic coupling ($g = 0$). For comparison, the lower panel shows the classical simulation (using $\Delta t = 0.5$ fs). We observe both slow and fast modulations of the population in time. The slow modulations stem from the superposition

of a pair of eigenstates of the electronic Hamiltonian $\hat{H}_{el}$ that arise from the mixing of donor-acceptor states when the donor is resonant with an eigenstate of the acceptor subsystem, as schematically shown in Fig. 2b. The fast modulations correspond to an energy gap around $\Delta_e$, indicating a contribution of the ground state near $-V$ to the superposition. We refer to Supplementary Notes for a detailed analysis of the states participating in ET.

In Fig. 3b we show $P_{transfer}(\Delta, T)$ resulting from simulations with different $\Delta$ after $T = 75$ fs and $T = 200$ fs (18 and 50 time steps). We compare quantum simulations without vibronic coupling ($g = 0$ in Eq. (4)) using 5 qubits for the sites and simulations incorporating vibronic coupling ($g = g_0$) to local oscillators using 5+5 qubits for sites and oscillators. The structure of $P_{transfer}$ is well reproduced and the peaks are revealed at the expected driving forces. Notably, the quantum hardware is able to resolve the vibronic resonance at $\Delta_v = 3010$ cm$^{-1}$ for the case of vibronic coupling ($g = g_0$), which only starts to build up after a few vibrational cycles of $\approx 22$ fs and continues to rise within the lifetime of vibrational excitations. This is evident when comparing $P_{transfer}(75$ fs$)$ with $P_{transfer}(200$ fs$)$. Depolarising noise tends to equilibrate the populations of all qubits and thus artificially raises $P_{transfer}$. This effect is most prominent in regions where neither electronic nor vibronic resonances are present[65], leading to an overestimation of $P_{transfer}$, as seen for example around $\Delta = 2000$ cm$^{-1}$ in $P_{transfer}(200$ fs$)$. The existing noise in the quantum hardware therefore sets a limit to the maximum time that one can integrate the probability of transfer while revealing structure, rather than adding a global shift in $P_{transfer}$. The electronic resonance shown in Fig. 3a is marked with a red cross at $\Delta_e$, while the vibronic resonance case shown in Fig. 3c is indicated in red markers at $\Delta_v$.

In Fig. 3c we show the population dynamics $P_n(t)$ of the quantum simulations with driving force $\Delta_v$ and compare them in the bottom panel to classical simulations ($\Delta t = 0.5$ fs, $N_b = 5$) with a damping rate of $(112.5$ fs$)^{-1}$, which we identified as the effective damping rate $\Gamma_{QC}$ (see Supplementary Fig. 4). The dynamics showcases the slow build-up of populations at sites $n = 2$ and $n = 4$ in the case of vibronic coupling (right), while in the case of no coupling (left) the excitation stays localized at the donor. This confirms that vibronic coupling is behind this mechanism by entangling electronic eigenstates with oscillator excitations. Maintaining the contrast in $P_{transfer}$ between $g = 0$ and $g = g_0$ at vibronic resonances is the basis for our proposal to use microscopic ET as an application-based benchmark.

In Fig. 3d we show the evolution of populations $P_n(t)$ at the vibronic resonance $\Delta_v = 3010$ cm$^{-1}$ in the presence of vibronic coupling ($g = g_0$). We show the raw experimental data (black crosses) and after error mitigation (orange circles), demonstrating the impact of our model-specific mitigation protocol (see Methods). While the raw data exhibits a fast equalization of the populations of all qubits, error-mitigated data is well matched to classical simulations. We show three classical simulations with increasing damping rates, where the highly damped simulation shows a stronger increase of the trap population, while the weakly damped simulation has a more effective electron transfer to the acceptor sites. The periodic pumping of population from the donor to acceptor sites gives rise to a *ratchet*-like growth of the populations $P_2(t)$, $P_3(t)$, $P_4(t)$, along with the quenching of population on the acceptor trap $P_1(t)$. These are signatures of a quasi-stationary, nonequilibrium state that provides ultrafast vibronic ET. Interestingly, the populations $P_n(t)$ of the sites with the largest contribution to this vibronic resonance ($n = 2$ and $n = 4$) are better reproduced than the population of *spectator* sites like $n = 3$, whose population stays relatively low and it is overestimated on the quantum computer (see Supplementary Notes for the involved eigenstates).

## Scaling Up the Number of Sites

In this section we probe the capacity of the quantum computer to scale up the problem size $N$ by simulating vibronic resonances from $N = 3$

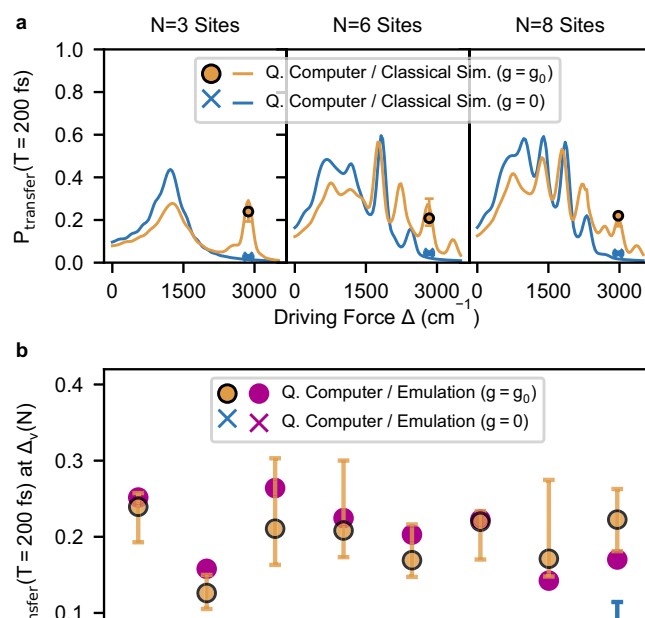

**Fig. 4 | Probability of vibronic ET for increasing number of sites.** For each system size $N$ we show experimental results for $P_{transfer}(T = 200$ fs$)$ obtained from 10 different quantum simulations on IBM_AACHEN for an ET-model with vibronic coupling ($g = g_0$) on up to 20 qubits, and 3 different quantum simulations without vibronic coupling ($g = 0$) on up to 10 qubits. The orange circles (blue crosses) show the median transfer probability for $g = g_0$ ($g = 0$), while the error bars indicate the smallest and largest value. **a.** We show the experimental results for $N = 3$, 6 and 8 over the driving force $\Delta$, together with classical simulations ($\Delta t = 0.5$ fs, $N_b = 5$). The vertical lines in magenta highlight the vibronic peaks which we chose for our quantum simulations. **b.** We show experimental results over the number of sites $N$, compared against noiseless emulations ($\Delta t = 4$ fs, $N_b = 2$) of the undamped model. Note that the height varies between vibronic peaks due to the changing energy landscape.

sites up to $N = 10$ on IBM_AACHEN. These simulations of ET pose an application-based benchmark, because the vibronic mechanism of ET relies on non-separable states where electronic sites and oscillators are entangled (see Supplementary Notes), and test the ability of the quantum processor to maintain entangled bipartitions of sites and oscillators over a considerable number of time steps. We simulate an ET-model with vibronic coupling ($g = g_0$) on 10 different days using $N + N$ qubits and compare against an ET-model without vibronic coupling ($g = 0$) at the same driving force, where ET is suppressed, on 3 different days using $N$ qubits. The constant depth of the Trotter circuits with respect to the number of sites $N$ thereby ensures comparability (see Discussion).

In Fig. 4 we show experimental results for the transfer probability $P_{transfer}(\Delta_v(N), T)$ after $T = 200$ fs (50 time steps) for system sizes $N = 3 - 10$ using up to 20 qubits. For the ET-model with $g = g_0$ ($g = 0$), the orange circles (blue crosses) give the median value for $P_{transfer}$ of the 10 (3) simulations while the error bars indicate the lowest and highest result. As shown in Fig. 4a, the energy landscape and the structure of vibronic resonances change with the problem size $N$, requiring us to carefully select for each $N$ a driving force that corresponds to a genuine vibronic resonance. The specific driving forces $\Delta_v(N)$ for each $N$ are given in Supplementary Notes.

In Fig. 4b we compare the experimental results for different system sizes $N$ against noiseless emulations. For every case, we obtained at least one run with very strong agreement to the emulation when using $N + N$ qubits in the presence of vibronic coupling ($g = g_0$). In the absence of vibronic coupling ($g = 0$), the probability of transfer at $\Delta_\nu(N)$ gets suppressed, and the experimental values using $N$ qubits lie even closer to the emulated values. The variance of experimental results across independent runs is significantly larger for $g = g_0$ ($N + N$ qubits) than for $g = 0$ ($N$ qubits). We attribute this to the availability of a sufficiently large set of connected qubits with low error rates, which is fluctuating between days for the used device. This is also illustrated by the outlier for $g = 0$ and $N = 10$, which compared to the simulations for $N \leq 9$ from the same day had to use one additional qubit with considerably worse error rates.

Furthermore, our simulations show good separation between the cases $g = 0$ and $g = g_0$ for all system sizes $N$, verifying that the increase in $P_{\text{transfer}}$ was indeed mediated by vibronic coupling and entangled site-oscillator states in the quantum processor, instead of being a product of depolarising noise.

To investigate the accuracy of our quantum simulations and to attribute an effective damping rate $\Gamma_{\text{QC}}$, we define the time-averaged deviation

$$\epsilon(T) = \sum_t \frac{\sqrt{\sum_n |P_n(t) - P_n^{(\text{ref.})}(t)|^2}}{T}. \tag{10}$$

calculated with a reference method, either emulations ($\Delta t = 4$ fs, $N_b = 2$) or classical simulations ($\Delta t = 0.5$ fs, $N_b = 5$). The deviation function $\epsilon(T)$ strongly depends on the value $\gamma$ used in reference calculations, and this dependence becomes more prominent for times $T \gtrsim \gamma^{-1}$. A high $\epsilon(T)$ relative to $\gamma = 0$, for example, does not necessarily indicate poor performance, as this deviation can be significantly reduced when comparing against reference calculations with finite damping ($\gamma > 0$). First, we compare our experimental results on the quantum processor to undamped ($\gamma = 0$, Fig. 5a–c) reference simulations, providing a clear benchmark that quantifies the hardware's capability to simulate long-lived, weakly damped dynamics ($1/\gamma \gtrsim 500$ fs), a regime that remains challenging for classical simulations[5,47]. Furthermore, we compare experimental results against damped ($\gamma > 0$, Fig. 5d–e) reference simulations, which allows to determine the effective damping rate $\Gamma_{\text{QC}}$ imprinted onto the pseudomode ET-model.

In Fig. 5a we show the spread of $\epsilon(T)$ values at $T = 200$ fs (50 Trotter steps), for each $N$ across 10 independent runs, when compared to emulations ($\Delta t = 4$ fs, $N_b = 2$) of the vibronic ET-model ($g = g_0$) without damping ($\gamma = 0$). While the deviation clearly increases as a function of the system size before error-mitigation (black crosses), discarding shots from the raw measurement results that violate the conservation of site excitations ($\hat{H}_{\text{el}}$, blue markers) allows us to reduce the median deviation below 10% to a value that is approximately constant for $N = 3 - 7$ sites. In addition, when we discard shots with more than a single vibrational excitation ($\hat{H}_{\text{el}}$, $\hat{H}_{\text{vib}}$, orange markers), the time-averaged deviation $\epsilon(T)$ stays now below 10% for all cases $N = 3 - 10$. Remarkably, the lowest deviation $\epsilon(T)$ achieved for any of these simulations is for $N = 7$ (7+7 qubits). This finding is in accordance to the hardware calibration data, as the qubit properties over multiple days are found to be best on average for $N = 7$. Besides, the experimental run for $N = 7$ with the lowest deviation corresponds to the experiment with the best qubit properties, in particular the highest minimum $T_1$, $T_2$ times and lowest gate errors (see Supplementary Fig. 6).

In Fig. 5b we show the same analysis when comparing against classical simulations ($\Delta t = 0.5$ fs, $N_b = 5$) with a shorter time step and up to 5 levels per oscillator. This comparison allows us to assess the impact of the Trotter error and the truncation of oscillators in our experiments with respect to a classical simulation, where both errors are negligible and can be considered numerically exact. While the Trotter error introduces an overall shift to $\epsilon(T)$ that is approximately constant for all $N$, truncation errors from oscillators are more pronounced for small system sizes ($N \leq 6$). Note that the error mitigation, which discards shots with more than a single vibrational excitation (orange markers), leads to a higher deviation for the smallest system size $N = 3$, as processes with two vibrational excitations have a non-negligible contribution to vibronic ET (see Supplementary Notes for a separate analysis of Trotter and truncation errors).

In Fig. 5c we plot the time evolution of the time-averaged deviation $\epsilon(T)$ with respect to emulations, showing for each system size $N$ and time point $T$ the fully mitigated experimental data with the lowest deviation. Surprisingly, the deviation does not increase with the system size $N$. This is consistent with the fact that we obtained for every $N$ at least one very accurate simulation, and with our observation that the accuracy of quantum simulations only depends on the availability of a sufficiently large connected set of good qubits. We observe slowly increasing deviation from the undamped dynamics as the simulation time increases, reaching a time-averaged deviation of 10% after around 300 fs, indicating the current limit in simulation time.

In Fig. 5d we assess the effective damping rate $\Gamma_{\text{QC}}$ of the quantum simulation with $N = 7$ sites from May 28 which is the run with the overall lowest deviation shown in Fig. 5b. To do so, we compare that quantum simulation to classical simulations ($\Delta t = 0.5$ fs, $N_b = 5$) of the target model with different fixed damping rates $\gamma$. We observe a minimum of $\epsilon(T)$ at $\Gamma_{\text{QC}}^{-1} = 62.5$ fs, for which the value of the time-averaged deviation does not increase further with $T$, i.e. $\epsilon(T) \approx \epsilon(T + \Delta t)$, indicating a successful reproduction of pseudomode damping. In Fig. 5e we show a quantum simulation with $N = 7$ from a different day (May 07). Compared to the May 28 run, the deviation $\epsilon(T)$ is generally higher and keeps increasing with no clear minimum at any damping rates $\gamma$. Overall, this indicates a larger amount of noise that is incompatible with pseudomode damping and overshadows the effect of an effective damping rate $\Gamma_{\text{QC}}$. The two simulations shown here are among those with most and least favorable average two-qubit gate fidelities and minimum $T_1$ times on the quantum circuit (see Supplementary Fig. 6). In summary, we find that the effective damping rates $\Gamma_{\text{QC}}$ vary substantially between different runs of the same experiment, and are mostly determined by the quality of entangling gates and the lowest $T_1$, $T_2$ times of qubits.

## Amount of shots required for error mitigation

Since we apply error mitigation as a post-processing method by discarding shots from the raw measurements (see Methods), the number of shots left for the computation of expectation values is an important metric. We discard shots that do not comply with particle conservation in the electronic subsystem ($\sum_{n=0}^{N-1} \hat{a}_n^\dagger \hat{a}_n = 1$ in $\hat{H}_{\text{el}}$). Furthermore, we set a limit to the number of vibrational excitations ($\sum_{n=0}^{N-1} \hat{b}_n^\dagger \hat{b}_n \leq N_{\text{max}}$ in $\hat{H}_{\text{vib}}$). While we chose $N_{\text{max}} = 1$ for all results presented above, the analysis of remaining shots in this subsection here chooses $N_{\text{max}} = 2$, motivated by the fact that states with two vibrational excitations participate to a small extent in the charge transfer process for the smallest system sizes ($N = 3$ or $N = 4$), ensuring that all discarded shots are truly stemming from errors. Nonetheless, it is worth noting that the choice of $N_{\text{max}}$ depends on the structure of the vibrational environment and the energy of the initial state.

To estimate every population $P_n(t)$ with the same accuracy $\epsilon_{\text{shot-noise}}$, the number of shots scales logarithmically with the size of the system $N_{\text{shots}} \sim \log(N)/\epsilon_{\text{shot-noise}}^2$ [66]. This scaling behavior stems from the statistical nature of quantum measurements (shot-noise) and is already present for ideal quantum computers without noise. In the presence of noise, more shots are demanded to compensate for the loss of incompatible shots after error-mitigation. To evaluate the rate

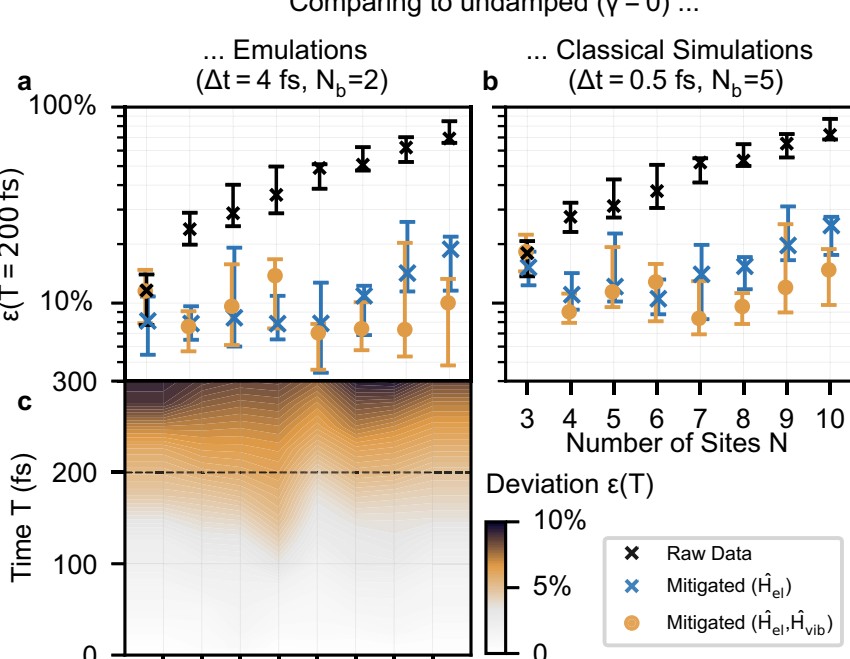

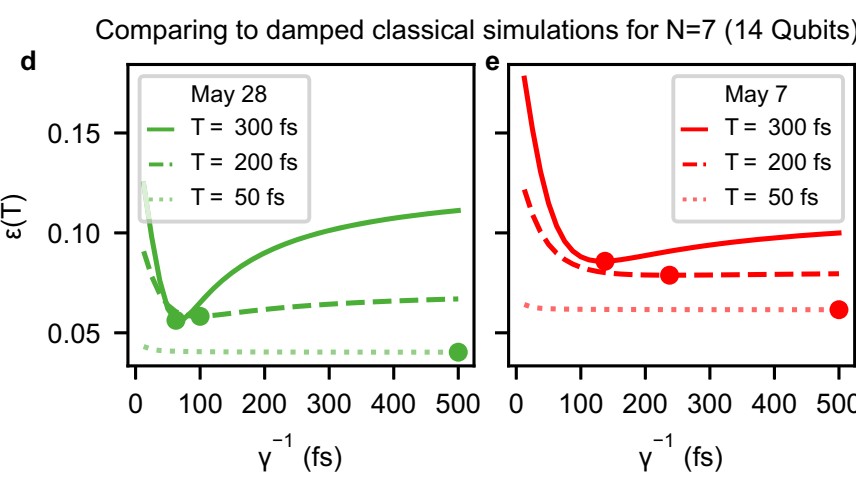

**Fig. 5 | Time-averaged deviation $\varepsilon(T)$ of quantum simulations of the ET-model with vibronic coupling ($g = g_0$) from multiple reference calculations with different damping rates $\gamma$. a** Quantum simulations at different levels of error mitigation (see Methods) compared against undamped emulations ($\gamma = 0$, $\Delta t = 4$ fs, $N_b = 2$ levels per oscillator) at $T = 200$ fs. The minimum, maximum and median of 10 simulations across different days is shown for each $N$. **b** Comparison against undamped classical simulations ($\gamma = 0$, $\Delta t = 0.5$ fs, $N_b = 5$) for $T = 200$ fs. **c** Heatmap showing the time-averaged deviation $\varepsilon(T)$ compared against the undamped emulations, as a function of the total evolution time $T$ of the fully mitigated results, whereby for each $N$ and $T$ the run with the lowest deviation is shown. The time corresponding to the errors shown in a and b is indicated with a dashed line. **d, e** Extraction of $\Gamma_{QC}$ by comparing two quantum simulation for $N = 7$ sites from different days against classical simulations with $\gamma$ ranging between $(12.5\,\text{fs})^{-1}$ and $(500\,\text{fs})^{-1}$. The dot markers indicate the minimum of each curve.

at which shots are lost due to errors in the hardware, every circuit was sampled with the same number of shots ($N_{\text{shots}} = 10^4$), regardless of its depth and number of qubits.

The number of remaining shots $S_N(t)$ as a function of the number of time steps is shown in Fig. 6a. We also plot in dashed lines the number of shots $S_{N,\text{mixed}} = N(1 + N + \binom{N}{2})/4^N$ that would remain if the quantum processor was in the completely mixed state, i.e. after loosing all quantum information due to noise. Expectedly, the number of remaining shots $S_N(t)$ decays toward this bound for every system size

$N$. It is worth noting that after 100 time steps, $S_N(t)$ remains above the completely mixed state $S_{N,\text{mixed}}$ in all cases.

A possible explanation for the loss of shots with increasing number of time steps are the different types of noise occurring on the hardware. Over the first time steps, we find a power-law behavior that we attribute to gate infidelities. This power-law decay is stronger for more qubits, corresponding to the increasing average gate errors as shown in Supplementary Table 1. After enough time steps, we find that the loss of shots transitions from power-law to exponential. This happens when the execution time of the circuit on the hardware

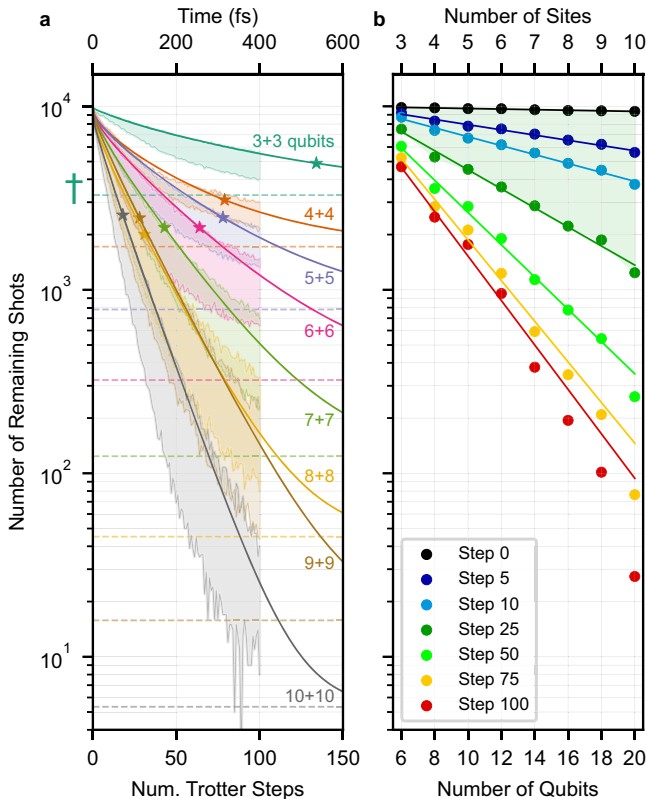

**Fig. 6 | Number of remaining shots $S_N(t)$ after error mitigation. a** Number of remaining shots as a function of the number of Trotter steps (1 step ≡ $\Delta t$=4 fs), after discarding measurements corresponding to unphysical states with more than two vibrational excitations. For each system size $N = 3 - 10$, the shaded area indicates the highest and lowest number of shots among the ten independent simulations with vibronic coupling ($g = g_0$). The dashed horizontal lines refer to the expected number of shots if the processor is in the completely mixed state, marked with a dagger for $N = 3$ (6 qubits). The solid lines correspond to a numerical fit (Eq. (11)), interpolating between a power law and an exponential decay towards the completely mixed state, with this transition indicated with star markers. **b** Remaining number of shots as a function of the number of qubits evaluated at increasing number of Trotter steps (0 – 300 fs). The solid lines correspond to a numerical fit with an exponential decay (Eq. (12)). The green area indicates the largest Trotter step for which the total execution time lies within the lowest $T_1$, $T_2$ times of qubits in the circuit ( ~ 30 $\mu$s for $N_{qubits} \leq 16$ and ~ 10 $\mu$s for $N_{qubits} = 20$).

approaches the regime of the smallest $T_1$ and $T_2$ times involved in the circuit, indicating the effect of continuous noise. To test this claim, we fitted for each $N$ the simple model

$$S_N(t; \tau, a, b) = S_N(t = 0)p_{valid}(t; \tau, a, b) + S_{N, mixed} \, p_{mixed}(t; \tau, a, b),$$
$$p_{valid}(t; \tau, a, b) = (1 - at^b)e^{-t/\tau}, \quad (11)$$
$$p_{mixed}(t; \tau, a, b) = 1 - e^{-t/\tau},$$

to the quantum simulation with the largest number of remaining shots $S_N(t)$ at the end of the simulation ($t = 400$ fs). Notably, the parameter $\tau$ (indicated with star markers in Fig. 6a) marks the transition from the power-law behavior of gate error propagation to the exponential scaling of damping and dephasing noise, and is indeed in good agreement with the lowest $T_1$, $T_2$ times when considering the circuit execution times in Supplementary Table 1. In particular, the transition occurs twice as late for $N = 3$, where no SWAP gates are required, reflecting that the corresponding circuit execution time is half as long as for $N > 3$.

In Fig. 6b we show the number of remaining shots as a function of the number of qubits at specific time steps. The solid lines at each Trotter step were obtained with a numerical fit

$$S_{Step}(N_{qubits}; C, r) = C \, e^{-rN_{qubits}} \quad (12)$$

with prefactor $C$ and rate $r$. While the number of remaining shots has an exponential dependence with the total number of qubits $N_{qubits}$, the rate $r$ is highly dependent on the number of time steps. For steps 0, 5, 10, 25, 50 and 75 we find $1/r$ values of 278, 31, 18, 8, 5 and 4 respectively, indicating at each Trotter step, the number of qubits for which the number of remaining shots is around $1/e$ of its initial value. The values of the prefactor $C$ in Eq. (12) stay close to the initial number of shots in our experiments ($10^4$) until 25 Trotter steps, while they almost double for 50 steps, indicating the scaling becomes worse than exponential after 25 steps ($T = 100$ fs). The execution time of circuits for 25 steps coincides with the lowest $T_1$, $T_2$ values in the circuit, illustrating again the importance of the minimum qubit coherence time.

To summarize, in the ideal scenario of a noiseless quantum computer, increasing the system size has only a logarithmic impact on the number of shots that are required to estimate individual qubit populations with the same level of accuracy. In the presence of noise, additional shots are needed to compensate for the loss of incompatible shots after error-mitigation. Our simulations reveal that the number of remaining shots has an exponential dependence on the system size $N$ with a small exponent, that increases for simulation times that exceed the coherence time of qubits in the circuit. Regarding the number of time steps, the loss of compatible shots scales polynomially within this coherence time, and becomes exponential for longer circuits.

## Discussion

This section investigates how far our quantum simulations can be extended to compete with classical simulations. First, we consider the quantum circuit corresponding to a single Trotter step, as outlined in Fig. 7a. The gates can be organized in four parallel layers, such that the circuit depth does not grow when increasing the number of sites $N$, enabling the simulation of arbitrarily large systems with near-constant quantum runtime. For system sizes with more than $N = 3$ sites (3 + 3 qubits), an additional layer of parallelized SWAP gates has to be introduced when mapping the model to the heavy-hex layout processor of IBM_AACHEN (see Fig. 1b for $N = 7$). While the mapping onto the heavy-hex topology is already very efficient, even shorter circuits with no SWAP gates and therefore longer time evolution are possible for quantum computers with a square topology.

To assess how the quantum approach presented here performs relative to state-of-the-art classical techniques, we compare to the tensor-network DAMPF algorithm[59], where computation time scales as $\mathcal{O}(N^3 M N_b^2 \chi^2)$ for a parallelized HPC-implementation of the 1D chain model as presented here with $N$ sites and $M$ oscillators per site, $N_b$ energy levels per oscillator and $\chi$ as the bond dimension of the matrix product operators, underlining that the main challenge for classical simulations lies in the scaling of the system size $N$. We note that while in general the complexity of DAMPF scales with $(\dim \hat{H}_{el})^3$, the single-excitation-manifold yields $\dim \hat{H}_{el} = N$.

In comparison, the number of qubits in our quantum algorithm scales linearly with the system size $N$, as illustrated in Fig. 7b with an efficient routing strategy for a square topology to scale up the number of sites to $N = 50$ using 100 qubits with $M = 1$, $N_b = 2$.

While the circuit execution time is constant with $N$, the current bottleneck lies in the amount of shots discarded in the post-processing of the error mitigation scheme. Extrapolating from our analysis in Fig. 6b, we expect simulations with 100 qubits over 100 timesteps will become feasible when gate fidelities and qubit coherence times have improved by another order of magnitude: Assuming hardware

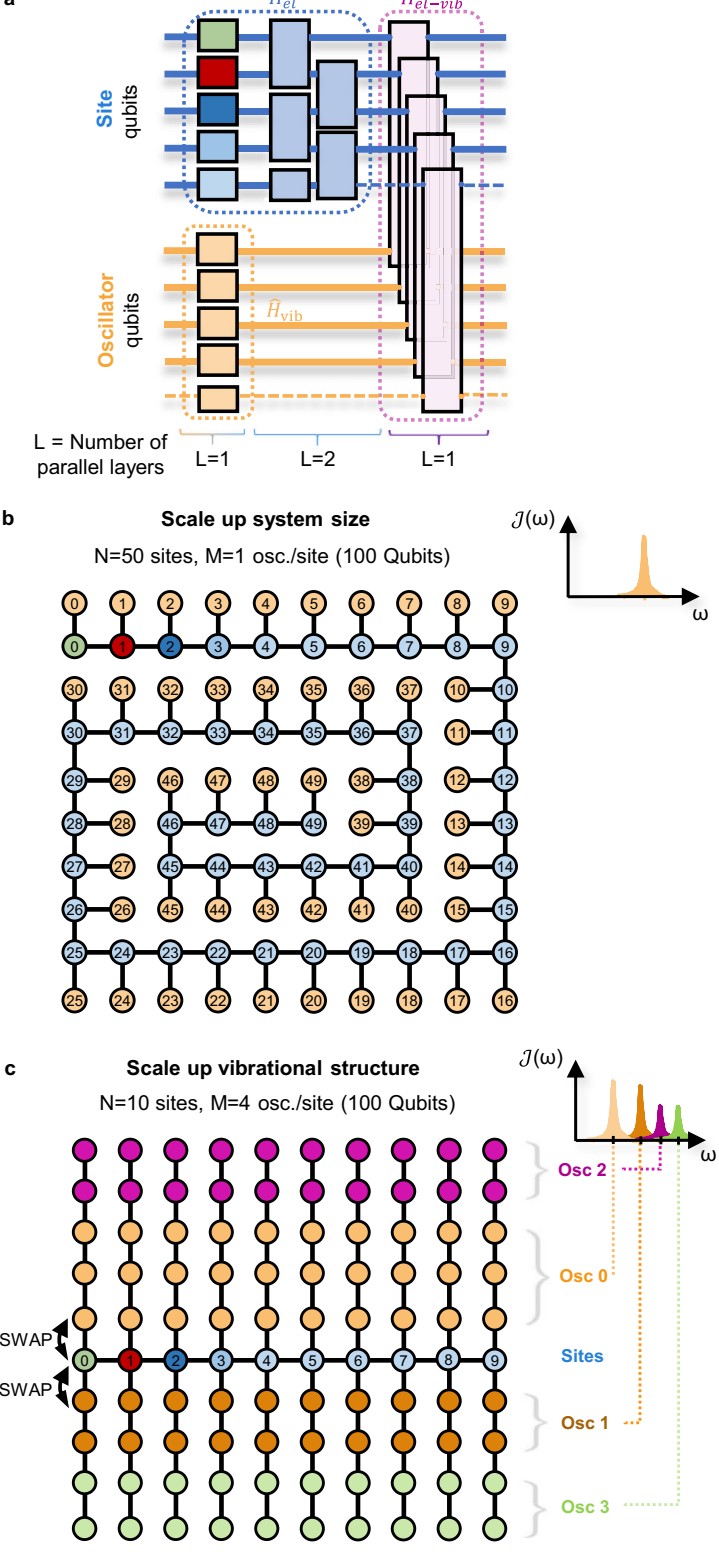

**Fig. 7 | Circuit structure and routing strategy for large-scale simulations on square qubit layouts. a** Quantum circuit corresponding to a single Trotter step, consisting of four parallelisable layers. The vibronic coupling term $\hat{H}_{el-vib}$ in magenta consists of parallelized two-qubit entangling gates between site qubits and their corresponding oscillator qubit. **b** Vibronic models with a single oscillator per site can be scaled up without increasing circuit depth. The example shows a model with $N = 50$ sites (100 qubits) on a $10 \times 10$ square layout processor. **c** When scaling up the vibrational structure by adding multiple oscillators per site, the local vibrational environment of site $n$ can be encoded in the upper/lower rows at the $n$-th column.

improvements lead to 10 times longer simulations with the same error and shot loss as today, we estimate using Eq. (12) that millions of shots will be required to simulate 100 time steps at a target accuracy of $\epsilon_{\text{shot-noise}} = 0.01$, which takes tens of minutes to sample at the high shot rates of superconducting processors, which sampled $10^4$ shots in our simulations within few seconds.

Although models with more complex vibrational structures are outside the scope of this work, it is also possible to include more oscillators and energy levels in our approach, as shown in Fig. 7c. Such multi-pseudomode models can be used to fit arbitrary spectral densities $\mathcal{J}(\omega)$, including features like thermal environments. If every site is coupled to independent vibrational environments, the number of qubits scales linearly with the number of oscillators $M$. When using a straight-forward encoding where each additional phonon level is mapped to a single qubit (see Supplementary Methods), the number of required qubits scales as $\mathcal{O}(NMN_b)$. More efficient encodings of oscillators could in principle lead to a logarithmic scaling in $N_b$. The quantum circuits can be parallelized with respect to $N$, similar to the Trotter circuit in Fig. 7a. Each oscillator qubit that couples to an electronic site adds an additional gate layer, causing the Trotter circuit depth to scale as $\mathcal{O}(MN_b)$. Note that this scaling holds for All-To-All, Square, and Heavy-Hex topologies. However, the latter require several layers of SWAP gates to move the system sites along the oscillator chains.

Significantly, classical simulations quickly become intractable in non-perturbative regimes where vibronic coupling is comparable to the strength of electronic interactions ($g_0 \sim J$) and oscillators are weakly damped ($\gamma \ll g_0, J$)[39,59], as exemplified in DAMPF, where the bond dimension that is required to stay below a certain error threshold is determined by the balance between the spread of quantum correlations (entanglement) and local dissipation[60]. In contrast, the quantum algorithm is independent of bond dimensions $\chi$ as the quantum processor naturally encodes the full vibronic state in the qubit-mapped Hamiltonian.

Beyond the model studied here, extensions of our approach to include arbitrary numbers of electronic excitations, or to include electron-hole dynamics with a second register of site qubits, promise an even larger speedup over classical methods like DAMPF, that scales with $(\dim \hat{H}_{\text{el}})^3$. Furthermore, our approach can be generalized to 2D lattice geometries, where the entanglement area law leads to even larger bond dimensions that challenge classical tensor network methods, while our quantum simulation does not suffer from this dimensional bottleneck, in particular when employing platforms with flexible coupling topologies: trapped ions with long-range connectivity and neutral atoms with reconfigurable geometries can accommodate 2D layouts. Similarly, shared bath models, where multiple electronic sites couple to a common environment, can be implemented efficiently on such platforms by routing several sites to the same oscillator, capturing environment-mediated correlations that are particularly costly to treat classically. More broadly, new quantum hardware designs make use of native bosonic elements with parametrizable damping rates, like the motion of trapped ions and superconducting resonators[34,35]. In particular, ion traps excel at capturing more complex vibrational environments[36,38,39].

In conclusion, the pseudomode approach opens an unconventional avenue in quantum computing. While here we have compared our approach to classical methods such as DAMPF, which can target arbitrary decay rates for each oscillator, the damping rates in superconducting processors are not freely tunable: they are determined by the intrinsic qubit dissipation, governed by hardware coherence times $T_1$, $T_2$ and the Trotter time step. For the target model considered here with equal damping rates on each oscillator, our qubit selection strategy addresses this constraint by prioritizing oscillator qubits with

similar and maximally long coherence times. The residual inhomogeneities in dissipation rates remain inconsequential for simulation times $T \lesssim 1/\Gamma_{QC}$, when the precise distribution of hardware damping rates is not yet resolved by the dynamics. Importantly, our quantum simulations were able to accurately resolve the peak structure of the time-averaged probability $P_{\text{transfer}}$, because the accuracy of individual populations are less relevant for spectral-dependent figures of merit like $P_{\text{transfer}}$, similar to the success of shadow spectroscopic techniques[67]. Long-lived correlations on the picosecond scale are industrially relevant for the design of new energy materials. While the quantum simulations presented here exhibit effective damping rates $\Gamma_{\text{QC}}$ in the order of $(100 \text{ fs})^{-1}$, we find the main bottleneck for large-scale simulations to be the limited number of qubits that are connected by high-fidelity gates and with sufficiently large coherence times. Simulating models with arbitrarily high damping rates is technologically feasible by artificially amplifying a particular type of noise[32,62], however the fundamental challenge still lies in the simulation of target models with lower damping rates. The simulation of vibronic ET for increasingly large problem sizes constitutes an application-based benchmark that, since the vibronic mechanism is driven by the entanglement between electronic sites and oscillators, measures the capacity of the quantum computer to sustain and scale up long-lived entangled states between well-defined bipartitions of the processor.

## Methods

### Pseudomode description of the vibrational environment

The width of a peak in the spectral density $\mathcal{J}(\omega)$ is inversely proportional to the lifetime of vibrational excitations (see Fig. 1a). This width also determines the duration of system-environment correlations as illustrated by the decay of the correlation function of the environment[68]

$$\mathcal{C}(t) = \int_0^\infty d\omega \, \mathcal{J}(\omega) \left( \coth \frac{\beta\omega}{2} \cos \omega t - \mathrm{i} \sin \omega t \right), \tag{13}$$

where $\beta$ is the inverse temperature (see Fig. 1a). Here we fix the temperature to zero, where the bath correlation function $\mathcal{C}(t)$ is the one-sided Fourier transform of the spectral density $\mathcal{J}(\omega)$. As shown in[47], thermal effects only become apparent after 150 fs, suppressing electronic resonances at high driving forces which require longer evolution to be resolved. In contrast, vibronic resonances are very robust against thermal effects, because they are driven by long-lived, intramolecular vibrations whose frequencies ($\omega_0 = 1500 \text{ cm}^{-1}$) are much higher than room temperature ($k_B T_{\text{room}} \approx 200 \text{ cm}^{-1}$). Incorporating finite temperatures in simulations is possible by introducing a low-frequency background that requires a higher number of pseudomodes.

A single broadened peak in the spectral density $\mathcal{J}(\omega)$, corresponding to a finite lifetime of vibrational excitations, can be captured in unitary dynamics using an infinite amount of oscillators ($M \to \infty$) around the frequency $\omega_0$ of the associated peak. An equivalent and simpler approach to reproduce such a broadened peak is achieved by coupling a single oscillator ($M = 1$, $b_n \equiv b_{n,1}$) with frequency $\omega_0$ to a secondary bath. Such a damped oscillator is called a *pseudomode*[53–57].

The resulting dynamics are governed by the following master equation for the density matrix

$$\frac{d\hat{\rho}}{dt} = -\frac{\mathrm{i}}{\hbar}[\hat{H}, \hat{\rho}] + \mathcal{L}_{\text{vib}}(\hat{\rho}), \tag{14}$$

where $\hat{H}$ is the Hamiltonian of the total vibronic system in Eq. (1) and $\mathcal{L}_{\text{vib}}(\hat{\rho})$ represents the Lindblad superoperator

$$\mathcal{L}_{\text{vib}}(\hat{\rho}) = \sum_{n=0}^{N-1} \gamma_n \left( \hat{b}_n \hat{\rho} \hat{b}_n^\dagger - \frac{1}{2} \{ \hat{b}_n^\dagger \hat{b}_n, \hat{\rho} \} \right), \qquad (15)$$

that describes the damping of the $n$-th oscillator due to its coupling to an auxiliary bath with a rate $\gamma_n$ (see Fig. 1b). As thermal effects are negligible, only amplitude damping towards the ground state is considered in Eq. (15).

After tracing out the vibrational degrees of freedom, this approach allows to engineer non-Markovian dynamics in the reduced state of the electronic system ($\hat{\rho}_{\text{el}} = \text{Tr}_{\text{vib}} \hat{\rho}$) in a non-perturbative manner. For a target spectral density $\mathcal{J}(\omega)$ of the physical model that one wants to simulate, the equivalence between the unitary dynamics of $\hat{H}$ with a continuous distribution of harmonic oscillators and the non-unitary dynamics of the pseudomode model with a single damped oscillator relies on matching the correlation function $\mathcal{C}(t)$ of the environment in Eq. (13)[53,54]. We note that an anti-symmetrized Lorentzian spectral function is required to satisfy the property of vanishing coupling at zero energy $\mathcal{J}(0) = 0$, which stems from thermodynamic constrains[68]. This would require using more than a single pseudomode to properly fit the anti-symmetrized Lorentzian. Nonetheless, when a single Lorentzian function is located at high-frequency, this only leads to negligible artifacts and electronic and vibronic resonances are still well resolved. We refer to Ref. 47 for a full analysis with numerically exact simulations.

## Quantum algorithm
We perform a Trotter expansion of the time-evolution operator of the full Hamiltonian in Eq. (1) over time $t_m = m \Delta t$,

$$e^{-i\hat{H}m\Delta t} \approx \left[ e^{-i\hat{H}_{\text{el}}\Delta t} e^{-i\hat{H}_{\text{vib}}\Delta t} e^{-i\hat{H}_{\text{el-vib}}\Delta t} \right]^m. \qquad (16)$$

For all our simulations on the quantum computer, we evolve an initial excitation in the donor site with a fixed time step $\Delta t = 4$ fs. This value represents a good compromise between total circuit depth and Trotter error and is motivated by the fact that the highest frequency in our Hamiltonian (Eq. (1)) is around $(11\,\text{fs})^{-1}$, stemming from the large Coulomb binding energy $V = 2420$ cm$^{-1}$ of bound electron-hole pairs in organic molecules. To validate the Trotter approximation, we compare to classical simulations with a smaller time step of $\Delta t = 0.5$ fs in Supplementary Notes.

We encode the Hamiltonian by mapping the $N$ nearest-neighbor coupled electronic sites on a chain of $N$ qubits, where measuring a qubit in state $|1\rangle$ means localization of an electron at the corresponding site. The $N$ quantum oscillators are truncated after the first excited state (involving $N_b = 2$ levels) and each oscillator is mapped onto a single qubit by identifying the number operator with the Pauli $\hat{Z}$ operator and the displacement operator with Pauli $\hat{X}$. The truncation is motivated by the fact that high-frequency modes cannot be thermally activated, and validated with simulations with $N_b = 5$ levels in Supplementary Notes. We refer to Supplementary Methods for the explicit qubit mapping, the involved gates and hardware execution times of the circuits. Note that the mapping presented there also allows for more levels per oscillator or the addition of more pseudomodes. Furthermore, we refer to Supplementary Fig. 5 for a comparison between simulations compiled with the variable-angle $R_X$ gate and the fixed-angle SqrtX gate as single-qubit gates.

## Exploiting hardware noise as computational resource
We assume that noisy processes in the hardware stem from two major sources. One noise channel contains amplitude damping and pure dephasing, which are continuously affecting all qubits, including idle

**Table 1 | Sources of hardware noise that are compatible with the pseudomode formalism**

| Lindblad op. $\hat{L}_{n,\alpha}$ | Type of Noise | Site qubits | Osc. qubits |
|---|---|---|---|
| $\hat{\sigma}_{n,\alpha}^-$ | Amp. Damping | × | ✓ |
| $\hat{\sigma}_{n,\alpha}^z$ | Pure Dephasing | × | ✓ |
| $\hat{\sigma}_{n,\alpha}^x, \hat{\sigma}_{n,\alpha}^y, \hat{\sigma}_{n,\alpha}^z$ | Depolarization | × | × |

Amplitude damping and pure dephasing acting on oscillator qubits can be exploited to engineer the target damping rate of pseudomodes. Noise acting on qubits that are used to encode electronic sites have no correspondence in our target model and hence need to be fully mitigated.

qubits. Another noise channel is depolarisation from gate noise. On the other hand, imperfections in the hardware implementation of single and two-qubit gates introduce additional detrimental effects such as depolarising noise that progressively destroys quantum coherence among qubits. Although we acknowledge there are other sources of noise, it is sufficient for our purposes to consider noise channels that can be described by Markovian theory. After mapping the full Hamiltonian Eq. (1) to operators acting on qubits in the processor, we model hardware noise with a master equation within the Lindbladian formalism

$$\mathcal{L}(\hat{\rho}) = \sum_{n,\alpha} \gamma_{n,\alpha} \left( \hat{L}_{n,\alpha} \hat{\rho} \hat{L}_{n,\alpha}^\dagger - \frac{1}{2} \{ \hat{L}_{n,\alpha}^\dagger \hat{L}_{n,\alpha}, \hat{\rho} \} \right), \qquad (17)$$

where the operators $\hat{L}_{n,\alpha}$ refer to noise processes acting on the $n$-th qubit, and the $\alpha$ index distinguishes between different noise processes. The core idea is to match the Lindbladian $\mathcal{L}_{\text{vib}}$ of the target open quantum system in Eq. (7) to the noise operators $\hat{L}_{n,\alpha}$. While this approach works generally on any type of quantum hardware, target models with pseudomodes are well suited to superconducting hardware due to the large availability of amplitude damping. As summarised in Table 1, we exploit the intrinsic amplitude damping of the qubits that are used to realize damped oscillators (pseudomodes) in Eq. (15) on the quantum processor. Because amplitude damping also introduces dephasing, pure dephasing acting on oscillator qubits can also be properly accommodated by considering an effective damping rate that depends on the $T_1$, $T_2$ times of each particular qubit. Furthermore, pure dephasing on qubits assigned to electronic sites can be identified with a constant offset in the spectral density[23], although this already constitutes a deviation from our target model. We refer to the Supplementary Methods for more details. Note that other noise processes are detrimental and need to be mitigated (see next subsection).

For any experiment of Trotterized dynamics on the quantum computer, the correspondence between hardware error rates and the effective damping rate $\Gamma_{\text{QC}}$ that one is able to simulate depends on the hardware execution time $T_{\text{exec}}^{\text{Trot.}}$ of one Trotter circuit and the particular choice of simulation time step, which we fixed to $\Delta t = 4$ fs for all our experiments on the quantum computer. For our simulations of the ET-model, a reasonable lower bound for the effective damping rate $\Gamma_{\text{QC}}$ is given by

$$\Gamma_{\text{QC}} \lesssim \frac{T_{\text{exec}}^{\text{Trot.}}}{\Delta t} \frac{1}{\min_q(T_1^{(q)}, T_2^{(q)})} \qquad (18)$$

where $\min_q(T_1^{(q)}, T_2^{(q)})$ refers to the smallest $T_1$ or $T_2$ time among the qubits in the circuit. In the general case, the inequality reflects the fact that $\Gamma_{\text{QC}}$ is limited by the qubit in the chain with the strongest damping. This mirrors kinetic competition in our electron transfer model (depicted in Fig. 1a): productive vibronic separation and lossy trapping constitute parallel pathways, where the faster channel dominates the

quantum yield by outcompeting alternatives. Similarly, among independent decoherence channels acting on different qubits, the fastest decay governs $\Gamma_{QC}$, establishing a hardware-imposed ceiling on simulable vibrational lifetimes. Note that Eq. (18) becomes equal for the case of equal damping rates on each qubit, while the exact value of $\Gamma_{QC}$ depends on the full distribution of $T_1, T_2$ times. This bound represents a qualitative statement, as it disregards other sources of noise in the quantum computer.

For selecting suitable hardware qubits in the processor we use Mapomatic[69], where most emphasis is given on finding configurations of qubits with the lowest amount of gate errors (single and two-qubit entangling gates) and low readout error. Among such configurations, we choose $T_1, T_2$ times that are as large as possible. We reserve the best qubits for encoding electronic sites, since only noisy oscillator qubits are compatible with our model (see Table 1). Nonetheless, we find that gate fidelities and the minimum of the $T_1, T_2$ times are the main limiting factor for accurate simulations when encoding either sites or oscillators in the quantum processor. We refer to the Supplementary Methods for further details about our criteria to select suitable qubit layouts.

### Error mitigation
To enable accurate simulations over many time steps while retaining a meaningful damping rate, we need to filter out other sources of noise that are incompatible with the target model insofar as possible. Entangling gates in superconducting processors can introduce errors such as strong depolarising noise that tends to equilibrate the populations of all qubits. We employ two levels of a model-specific error mitigation scheme, that is hardware- and implementation-agnostic and can partially correct for depolarizing errors and amplitude damping on electronic sites (see Table 1). It is complemented with readout error mitigation[70] for the results presented in this work.

First, we discard shots that do not comply with particle conservation in the electronic subsystem ($\sum_{n=0}^{N-1} \hat{a}_n^\dagger \hat{a}_n = 1$ in $\hat{H}_{el}$), since the creation or annihilation of electronic excitations is not possible in the Hamiltonian model (Eq. (1)). This particle conservation constitutes a powerful error mitigation strategy at the post-processing stage that is crucial for long-time evolution[62].

Secondly, we set a limit to the maximum number of vibrational excitations ($\sum_{n=0}^{N-1} \hat{b}_n^\dagger \hat{b}_n \leq 1$ in $\hat{H}_{vib}$), as the total energy in the system is usually around at most $2\omega_0$ and spreads over all electronic sites and all oscillators. As schematically shown in the right panel of Fig. 2b, vibronic resonances are driven by the entanglement between electronic states and quantized vibrations, forming non-separable superpositions of zero and one-phonon configurations. For this reason, we only keep shots with a small number of vibrational quanta. Note that the number of vibrational quanta should not be limited for models with oscillators that can be thermally activated, nevertheless, extensions to the pseudomode formalism with coupled oscillators can accurately reproduce nonzero temperatures while initialising all oscillators to their ground state, regardless of their frequencies[53].

When combined with noise characterization and error mitigation techniques like randomized compiling and probabilistic error cancellation, partial control over error rates in the qubits could be achieved[26]. However, we note that since one goal of this work is to maximize vibrational lifetimes and maintain system-bath coherence as long as possible, we did not apply such mitigation techniques that would address coherent gate errors.

### Data availability
Source data underlying all figures and tables are provided with the paper. The raw data are available under restricted access for institutional property regulations. They can be requested from the corresponding authors for verification and reproducibility purposes,

subject to institutional approval. Source data are provided with this paper.

### Code availability
The code used to generate the data for this study is partly based on proprietary software developed at HQS Quantum Simulations GmbH and the German Aerospace Center (DLR). Due to institutional intellectual property restrictions, this part of the code cannot be made publicly available. Non-proprietary components and example scripts are available from the corresponding authors along with the raw data.

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

## Acknowledgements

We thank Elias Walter and Arta Schellhorn for helpful discussions.

## Author contributions

A.D.S. and M.G. conceived the idea and developed the model. M.G., A.D.S., P.S., and M.M. build the methodology for simulations on the quantum computer. M.G. performed the experiments and emulations. A.D.S. performed classical simulations. M.G., A.D.S., and G.S. analyzed the results and wrote the original draft. M.M. and B.H. supervised the project and revised the manuscript. B.H. was responsible for the acquisition of funding. All authors discussed the results and reviewed the final manuscript.

## Funding

This project was made possible by the DLR Quantum Computing Initiative and the Federal Ministry for Economic Affairs and Climate Action; M.G., A.D.S., P.S., M.M. and B.H. were supported by BASIQ (qci.dlr.de/projects/basiq), G.S. by ALQU (qci.dlr.de/projects/alqu), and M.G., A.D.S. and G.S. by the project ELEVATE (Enhanced Problem Solving with Quantum Computers). M.G., A.D.S and B.H. further acknowledge support for the following points: From the state of Baden-Württemberg (KQCBW) for the use of the ɪʙᴍ_ᴀᴀᴄʜᴇɴ quantum computer. For classical simulations, the usage of the JUSTUS 2 cluster, supported by the state of Baden-Württemberg through bwHPC and the German Research Foundation (DFG) through grant no. INST 40/575-1 FUGG. For emulations, scientific support and HPC resources were provided by the German Aerospace Center (DLR). The HPC system CARA is partially funded by "Saxon State Ministry for Economic Affairs, Labour and Transport" and "Federal Ministry for Economic Affairs and Climate Action". Open Access funding enabled and organized by Projekt DEAL.

## Competing interests

The authors declare no competing interests.
