## [Transparent Peer Review file · Nature Communications]

Simulating Electron Transfer on Noisy Quantum Computers

Corresponding Author: Professor Birger Horstmann

Version 0:

Reviewer comments:

Reviewer #1

(Remarks to the Author)

The authors propose using the pseudomode formalism to simulate electron transfer in the presence of nucleon vibrational degrees of freedom as a non-Markovian bath and demonstrate the simulation protocol on an IBM QPU. The pseudomode-based digital-analog hybrid approach enables demonstrations to be executed on cloud QPUs without the extensive calibration information required by approaches that rely on residual ZZ interactions. While it is unclear whether this approach is scalable to a quantum utility level, the demonstration of simulating open-electron-transfer dynamics and the approach potential in the NISQ regime are worth publishing. I recommend that this manuscript be published after considering the following comments.

Major comments:

As the authors noted, the 1D dissipative model with the pseudomode approximation is well-suited to classical tensor network simulation. It seems unlikely that increasing the chain size as suggested in Fig. 3b would provide a favorable scaling for the quantum simulation over the class algorithm. It would be helpful if the authors could suggest in the manuscript under what circumstances their approach could potentially provide an advantage over the classical one.

While the authors suggest they present a digital quantum simulation toolbox, their approach appears to be a digital-analog hybrid, as they lack control over the oscillator qubit's dissipation. Beyond this terminology issue, several related aspects require further clarification.

- How do the variations of T1 and T2 among the oscillator qubits vary after the simulation? While the authors suggest that the worst qubit dominates the effective damping, as shown in equation 13, don't the variations also play an important role? It seems unlikely that the two cases (a) all oscillator qubits have poor T1 and T2, and (b) only the last oscillator qubit has poor T1 and T2, will give similar results.

- How is g being chosen for the $g > 0$ case, and what is its value used in the experiment and simulations?

- In section IIIB2, the authors state that a target damping rate $\gamma = (1\text{ps})^{-1}$ is used to compare the experimental results.

However, in Appendix D, the experimental results are compared with simulation results over a γ range of $(12.5\text{fs})^{-1}$ to $(500\text{fs})^{-1}$, and the best-fitting damping rate is $\sim (100\text{-}200\text{fs})^{-1}$. Does it mean that the performance measured in Fig.6 has a weak dependence on γ for an overall error level of 10%? If so, it seems odd since γ plays an important role in simulating the vibronic resonance.

Minor comments:

What is the justification to assume that each site is coupled to an independent vibrational environment? If the frequency ω_0 is associated with a specific set of intramolecular vibration modes, shouldn't the electron be coupled to the same set of modes at different locations in the potential well?

Table I suggests that the amplitude damping, pure dephasing, and depolarization of the site qubits need to be fully mitigated. In the experiment, is it correct that only the amplitude damping and depolarization are partially mitigated by the postselection, and the pure dephasing is not mitigated? Also, while randomized compiling and (partial) probabilistic error cancellation are mentioned, are they used in the experiment?

p.4: "The lifetime of vibrational excitations is inversely proportional to the width of the peak in the spectral density $J(\omega)$ (see Fig. 1 b)." It should probably be Fig. 1a instead.

p.5: "Our quantum circuits simulate the closed model in Fig. 1 a with $\gamma = 0$ using Trotterized quantum evolution on the quantum computer and 104 shots per time step." The simulations shown in part III use $g = 0$ instead of $\gamma = 0$.

p.6: "On the other hand, imperfections in the hardware implementation of single and two-qubit gates introduce depolarising noise that progressively destroys quantum coherence among qubits." The gate errors effectively reduce to depolarising noise only when randomized compiling is implemented. It is also technically incorrect to include the gate error in equation 12, since the effective depolarising noise is at the level of a randomized compiled Trotter step block rather than the continuous master equation evolution.

Reviewer #2

(Remarks to the Author)

In this work the authors perform a quantum simulation, using an IBM-Q cloud quantum computer, of electron transfer through a one-dimensional chain, where each site is coupled to local non-Markovian environment (which are included in the simulation via ancilla qubits). Applications to the study of light-harvesting and photo-cells are discussed.

The idea of simulating open-quantum systems on quantum computers is seeing growing interest, both for purposes like the authors explain in this work, and to more abstract problems, like finding many-body ground-states via dissipative state engineering. I found this article to be well written, and clearly explained, with a wealth of detail on the steps they made to implement their protocol that will surely be useful for future works.

The novelty in this work is largely in that detail, and in the simple but useful steps the authors make in improving and analyzing their results (like the error mitigation steps based on discarding shots that don't conform to physical expectations of their model, and using the intrinsic dissipation of the qubits themselves to get the correct ancilla environmental noise). On the positive side, their study implies larger scale simulations, at least of models similar to the ones they construct are feasible in the near future, and could be actually useful. On the negative side, the framework they build up is made of largely known concepts, so I felt, at least from the theoretical point of view, the impact might not be so strong. In addition, moving away from many of the model assumptions they make seems difficult, and no analysis of the impact of doing so is provided.

I have a few questions and comments that I hope the authors can consider which may help improve the work on these points:

- 1) Am I correct in understanding an important step is to make sure that the "worst" qubits are chosen for the pseudomode ancillas? And that, in the end, the 'simulation parameters' are essentially given by the hardware? There was some discussion I guess about expanding the run time of the circuit, to effectively increase that dissipation if needed, but this was not done in practice in any of the examples, is that correct?
- 2) I was wondering, is it given (1) is it possible to change the ancilla dissipation, without changing the "system" qubit dissipation as well? Or would one have to employ more sophisticated steps, like yet more ancillas + readout?
- 3) A minor comment, below equation 8 the authors state " We fix the temperature to zero, where the bath correlation function $C(t)$ and the spectral density $J(\omega)$ form a Fourier pair." While true for the very narrow Lorentzians ($\gamma \ll \omega_0$) the authors employ here, in general this is not the full story; as they mention at the end of the page, $J(\omega)$ is only defined for positive frequencies, so I believe extending the integral to negative frequencies requires taking this account.
- 4) Following on from this, the authors do not particularly explore issues arising from broad environments. They mention finite temperatures, and multiple Lorentzians, so in some sense the components one would need are there, but a bit more detail, or a recipe for general cases, would be useful. This problem has been explored, using pseudomodes, in dissipative state engineering applications, and ideas from those explorations seem useful here too (like using coupled pseudomodes, unphysical pseudomodes, etc), see e.g., <https://doi.org/10.1063/5.0283315> and <https://doi.org/10.1103/PhysRevResearch.6.043229>. It may also be good to cite the original pseudomode framework paper from Garraway and others B. M. Garraway, Phys. Rev. A 55, 2290 (1997), *ibid.* A 55, 4636 (1997), A. Imamoglu, Phys. Rev. A 50, 3650 .
- 5) The authors mention potential scaling upto 100 qubits given better gate fidelities in the near future; however, given the single excitation subspace their particular example employs, and the MPS approaches available to included the dissipative environment, could this case be said to eventually give rise to a model which is out of reach classically? A little bit more discussion on this would be useful.
- 6) Many of the technical implementation details are focused on the IBM-Q device they employ. But, the logic and approach seems generally useful for other devices. The text however often switches back and forth between generality and specificity. Given how soon particular devices become obsolete, and the pace of this technology, a small adjustment the presentation may make the results here more transparent and future-proof (e.g., presenting a recipe for analyzing and fixing errors in these types of devices, with the IBM results as a particular example). I think the components are there, it is just a small change in logical flow of the text.

Reviewer #3

(Remarks to the Author)

This article proposes protocols for the quantum simulation of open quantum systems and it presents specific results for non-equilibrium electron-transfer dynamics.

The general idea for the quantum simulation protocol is based on the pseudomode model, a theory used for the non-perturbative analysis of Gaussian open quantum systems.

It consists in replacing a Gaussian environment with a discrete set of lossy harmonic modes whose parameters are chosen in order to reproduce, or at least approximate, the environmental correlation function of the original bath. The authors consider different possible layouts in which the qubits representing the system are coupled to the qubits representing the environmental pseudomodes. Even within the limitation imposed by the interaction pattern in a quantum chip, the authors envisage the possibility to have multiple pseudo-degrees of freedom interacting with each of the system qubits using swap gates. Interesting extensions of this idea could involve actual harmonic modes to represent the pseudomodes, for example using trapped ions.

As a specific example, the authors consider the simulation of a model for exciton dissociation at an interface made by electron donor-acceptor materials. This model consists in a tight-binding in which the first site corresponds to the donor and the remaining sites to the acceptor. The on-site biases in the acceptor encode the effect of the Coulomb attraction between the electron and the hole in the donor, whose bias, or the detuning (driving force) with respect to the first site of the acceptor chain, is here considered as a free parameter.

The tight-binding model is further complemented by the presence of bosonic environments coupled to each of the sites to describe molecular vibrations. These environments are modeled using the pseudomode model. In particular, each environment is approximated using a single pseudomode, and further represented with a single qubit.

As shown in [76], which includes one of the authors, the charge transfer away from the interface can involve two different physical mechanisms, depending on the intensity of the driving force. For weak driving forces, the energy of the donor can be approximately on resonance with some of the extended eigenstates of the acceptor. The resulting hybridization can then lead to charge propagation away from the interface. While stronger driving forces are going to be detuned with respect to the eigenenergies of the acceptor, they can still be on resonance with "vibrationally hot" (i.e., including a vibrational excitation) delocalized states, leading to a "vibration-assisted exciton dissociation" [76]. Importantly, the phonons involved in this mechanism are characterized by a high frequency spectrum. While the description of a generic Gaussian environment might require several pseudomodes, in this case, the high-frequency characteristic of the spectrum makes it suitable for a single mode description. In fact, since each pseudomode is characterized by a Lorentzian spectral density, its peak at high-frequency mitigates nonphysicalities at zero and negative energies. I also note that, as shown in [76], the performance of the environment in assisting the propagation away from the interface degrades at intense damping, thereby further justifying the single-mode approximation in the case of a narrow spectral density.

In this context, this article reproduces some of the results of the classical numerical simulation reported in [76] by performing a quantum simulation using up to 20 qubits. The gates are defined through Trotterization of the dynamics and their error is modeled in terms of an effective Lindblad equation acting on both system and pseudomodes qubits. In this way, the noise affecting the environmental-qubits is also used to model the broadening of the pseudomode spectrum. To further improve the visibility of the data, the authors also considered a mitigation scheme in which the raw measurements results are post-processed to discard instances in which particles are not conserved in the system and in which the number of vibrational excitations exceeds a certain threshold.

The authors provide the results of different simulations, mostly focusing on the analysis of the time-averaged electron-transfer probability which is a function of the overall simulation time and the driving force. The results of the simulation are compatible with the main physical results mentioned above and obtained in [76]. Furthermore, the experimental data are compared to both classical emulation of the quantum circuit and a classical simulation following the numerical techniques introduced in [76].

Overall, I personally find the main topic of this article, i.e. the simulation of structured open quantum systems on a quantum computer, very interesting. The possibility to simulate quantum effects originating from the mediation of a quantum environment could constitute a very relevant and potentially impactful field of research for quantum technology. More specifically, the modeling of the dynamics in these exciton systems can be relevant to improve our main theoretical understanding on the mechanisms underlying charge propagation for potential technological applications in the development of solar cells.

The article is also, in my opinion, very well written. It describes the model, the simulation, and the results extremely clearly. The overall modeling of the circuit errors, including the qubit mapping and the comparison with emulation and simulation also witnesses a precise and methodical analysis whose results I have no reasons to doubt.

I would also like to mention that, in my opinion, the specific model for electronic transfer constitutes a very clever choice. In fact, on one hand, some physical effects of the model rely on system-bath hybridization to allow transitions to extended states, thereby justifying the focus on a quantum simulation. At the same time, the quantum environments considered here, can also be modeled using the simplest possible pseudomode model, i.e., one where each site is coupled to a single pseudomode. In other words, the authors considered an environment which has non-trivial effects on the system while also being simple to describe with pseudomodes, thereby allowing a quantum simulation within the current technological limitations.

While such a precise choice is evidence of the authors' knowledge, it also hints towards a rather specificity of these results,

which is in a bit of contrast with respect to the overall generality of the authors' narrative. More precisely, the authors mention that their goal is to "present a toolbox for digital simulation of large open quantum systems" and that this general approach is validated by studying the electron-transfer dynamics as a specific model. However, the choice of focusing on this specific model and dynamics struck me as very clever exactly because it tightly includes all the assumptions required for the simulation to work, so that it might be not so easy to go beyond it.

In other words, rather than presenting a toolbox with a specific application, I feel this article presents a very specific application with some further additional interesting ideas to move forward.

For example, the possibility to model each environment with a single pseudomode (further represented by a single qubit) is likely not going to hold in more general situations. In fact, even here, the parameters are such that the broadening of each mode corresponds to a regime which is close to the one defined as strongly damped in [76]. In this case, I imagine the single mode assumption might start to not be valid anymore. At the same time, the approximation of the harmonic Hilbert space in terms of a qubit is further justified by imposing a threshold on the pseudomode total population to 1, which might not be possible to impose in general situations. It is not really clear to me how to move beyond this without including new ideas.

It is important for me to mention that this is not a criticism with respect to the actual content of this article but, rather, just about the generality of the narrative. In fact, I do understand that, for example, the broadening of the modes is determined by the overall errors building in the simulation which is definitely going to be improved in the future generation of chips. I also understand that the authors do present interesting ideas in order to simulate multiple pseudomodes for each environment using swap gates or using "native bosonic elements" in other platforms. However, I have to note that, overall, the results of this quantum simulation are really a subset of those presented by the classical techniques in [76], which appear to me much more efficient. At the same time, as mentioned, the very possibility to even perform this simulation relies on choosing the model and its regime extremely carefully. While I do appreciate the ideas for further improvement, it seems to me that they will be very challenging to implement and, in any case, they go beyond the content presented here.

To summarize my point of view, I would like to simply take into account the title: "Simulating Electron Transfer on Noisy Quantum Computers: A Scalable Approach to Open Quantum System".

In my opinion, the content would be much better represented by the first half, i.e., "Simulating Electron Transfer on a Noisy Quantum Computer."

In other words, I have to highlight the clever specificity, rather than a broad generality, of the content presented. I further have to comment on the fact that, despite this specificity, the quantum nature of the simulation in this model is not, as far as I can see, close to the efficiency of the classical algorithms it was compared to.

At the same time, I personally found the content of this article very interesting, technically solid, and very well presented. I specifically appreciated the clever choice of this specific application alongside the careful analysis and interpretation of errors. The ideas for further extensions are, in my opinion, also very interesting. As a consequence, despite noting a lack of generality, I also believe this article will definitely be a source of inspiration for researchers in this field.

In summary, while I personally much appreciated the quality of this work, the main issue that prevents me from suggesting publication in Nature Communications in the present form is that its results are very specific, thereby not easily generalized into an actual toolbox for the broad community to use. Furthermore, within the specific domain considered here, the results of the quantum simulation are still quite far from the classical techniques which inspired it. As a consequence, I would like the authors to consider revising the generality of the overall narrative. Alongside this, I think it would be necessary to further add some estimates about the possible performance of the quantum simulation in more general regimes, such as to simulate an environment requiring several pseudomodes. One option in this direction could be presenting some estimation for the scalability of error mitigation when used to model several pseudomodes per qubit using the swap gates mentioned in the manuscript. Another option could be to estimate the performances for the simulation of actual harmonic modes, i.e., without the two-level approximation.

In my opinion, without such additional considerations, these results do not show evidence of the scalability mentioned in the overall title.

Below, I present a few specific considerations.

-) As mentioned, I really enjoyed reading about the idea for a simulation of several pseudomodes for each environment using the swap gates. However, it is not clear to me how the authors intend to further improve the simulation to actually recover Gaussianity. For example, even in the single pseudomode case considered here, could the authors present some ideas about a strategy to simulate higher excited states of the harmonic Hilbert space?

-) Some of the heatmap plots are used to represent the occupation probability at specific sites. Despite the discrete nature of the "position-label", the plots appear to be continuous. I understand that, in [76], this was explained as "the continuous nature of the contour plot emerges when displaying curves of equal populations that are extracted from the time-series data." However, I am not sure I really understand this point and I would then like to ask the authors whether they could add a comment explaining this feature in more detail.

-) As I am not an expert on the physics behind the charge transport, I would like to ask the authors whether the presence of an environment for each of the sites is necessary to justify the propagation away from the interface. For example, would the

same effect still be present in case only the first site of the acceptor was coupled to its environment. Is it necessary to couple all the sites to achieve the same transport results?

Version 1:

Reviewer comments:

Reviewer #1

(Remarks to the Author)

The authors addressed most of my comments made in my previous report in the revision. Nonetheless, the potential advantage of the proposed approach is unclear to me, as explained below. While I maintain my position that the demonstration and the potential of this approach in the NISQ regime make this manuscript worthy of publication, a discussion of generalizing this approach beyond 1D simulations would make it even more appealing. This manuscript will be suitable for publication in Nature Communications after the suggested change.

In the revision, the authors moderate their stance on their approach being a scalable approach for open quantum system simulation, and focus their discussion on the specific 1D example with an additional complexity analysis. While the complexity analysis is helpful for the readers to understand the potential of the approach, it is important to point out the following three points.

(a) For 1D systems, the advantage of the quantum simulation over the classical tensor-network simulation is a polynomial speedup in the system size N and unclear speedup related to the bond dimension of the tensor-network simulation. In my opinion, the $O(N^3) \rightarrow O(N)$ speedup is very unlikely to deliver any quantum utility in the NISQ and early fault-tolerant regimes, given the huge overhead required by QPUs. In the fully fault-tolerant regime, it is unclear, but the digital-analog hybrid approach is not targeted for that regime, nevertheless.

The authors mentioned that the bond dimension grows exponentially in the non-perturbative regime for the DAMPF method. (Exponential in time?) Is there a reference for this scaling? My understanding of tensor-network methods is that strong coupling does not necessarily mean they no longer work.

(b) It is also important to be aware that the comparison is not apple-to-apple, given that the analog part (ancilla qubit dissipation) is uncontrollable and may not even be completely known. Realistically, the qubit decay rates fluctuate and drift in time, and thus, even a complete calibration (if possible) would not allow the exact decay rate distribution at the time of the experiment to be known. While an upper bound on a relevant characteristic decay time can be estimated, the proposed approach cannot provide a quantitatively accurate result for a specific target set of decay rates. While the authors suggest that limited control is possible by engineering the total time and time step, this will not allow any control of the actual decay rate distribution. In this regard, the complexity analysis is unfair as the proposed protocol and the DAMPF method do not make the same calculation.

(c) In my opinion, it will be very helpful for the authors to make a brief discussion on whether their approach can be readily generalized to simulate 2D models, and to other noise models (maybe shared baths). This discussion will make the approach more toward a toolbox as proposed, rather than a simulation of a specific problem.

Reviewer #2

(Remarks to the Author)

I am satisfied with the authors' response. I think this article serves as a useful demonstration of how to perform quantum simulations of open quantum systems, and will be of great interest to the community.

Reviewer #3

(Remarks to the Author)

I would like to thank the authors for considering and replying to my comments. For example, I would like to start by further thanking the authors for updating the title which is now, in my opinion, a more precise representation of this work.

However, I have to mention that I still have some doubts about the generality of some of the overall claims. For example, even in the very abstract, reference to scalability is still present as

"our approach enables scalable simulations of non-Markovian dynamics on near-term quantum hardware."

Some of the comments provided in the reply helped me to understand the authors' point of view on this point a bit better. Citing from the reply:

"We emphasize that the primary challenge in simulating open quantum systems lies not in the

complexity of the vibrational environment per se, but in the scaling of the system size;"

and

"The goal of our manuscript was to explore increasing system sizes and the feasibility on real quantum hardware."

It seems to me that these sentences well summarize the point of view of the authors, mostly concerned about the scaling in terms of system size other than in the complexity of environmental effects. However, while it could be the case that the system size scaling is the primary challenge for this specific case, I would argue that it is not really true for general non-Markovian open systems.

To be more specific, in the comparison with the DAMPF algorithm, the scaling with respect to the system size is reduced from $O(N^3)$ to $O(N)$. While the scaling in DAMPF could be optimized to $O(N \log(N))$, this does represent an improvement, always justified by the direct use of quantum hardware to represent the closed system, fully in line with the stated authors' goal. Importantly, the authors do also mention an improvement in terms of environmental properties, i.e., the scaling with respect to the pseudomodes truncation N_b from quadratic to linear, since each of the energy levels is mapped to a single qubit.

However, while I can agree with the scalability in terms of the system size, the reliance on a very specific choice for the environment makes it difficult for me to justify the generality of the scaling in terms of environmental properties. In other words, the authors' goal about optimizing the scaling in terms of system size is, in my opinion, more suited to the analysis of quantum algorithms for closed systems rather than open, making it difficult for me to support the point of view behind the claim "our approach enables scalable simulations of non-Markovian dynamics".

Trying to be more specific, the reported estimates for the scaling in terms of environmental properties does rely on a specific model in which all the effects of the environment can be encoded using a single locally-dissipative harmonic mode for each system site. It is not clear to me that this scaling would still be valid for more general non-Markovian dynamics such as (i) deeper in non-perturbative regimes where the qubits non-linearity could hinder the validity of the effective model or (ii) in a regime relying on non-local coupling in the system-sites labeling (which might even hinder the scaling with respect to the system size) or (iii) for broader local environments (where $C(t)$ is not just the Fourier transform of $J(\omega)$).

While I understand that all of the above is beyond the scope of this work, my opinion is only intended to support my personal opinion that general claim about scalability might, possibly, be better to be toned down a little or postponed until fully supported by future analysis.

In summary, while I do appreciate the framework provided for the potential simulation of non-Markovian open quantum systems and the overall high quality of this work, I also do think that these results rely on a very specific choice of open quantum system. As a consequence, I would argue that the generality of the overall claims might also be better to be adjusted accordingly.

As a minor additional comment, I was wondering about the reply to referee 2 question 13, also related to my point (iii) above. As far as I know, $C(t)$ is not the Fourier transform of the physical spectral density $J(\omega)$ at zero temperature. I would then thereby want to ask the authors whether they could specify the precise meaning of the term "Fourier pair" used in the manuscript.

Version 2:

Reviewer comments:

Reviewer #1

(Remarks to the Author)

The authors addressed my concerns in the revision, and I support the manuscript being published in its current form.

Reviewer #3

(Remarks to the Author)

I would like to thank the authors for considering my suggestions and for updating the manuscript in a way that, in my opinion, more precisely represents the overall context of the reported results.

I would also like to thank the authors for the interesting comments about non-Markovian extensions given in their reply. The related added paragraph about possible advantages for the modeling of shared bath models describes, in my opinion, a rather intriguing feature for future developments.

Given the revised narrative and its overall high quality, this work could spark, in my opinion, the interest of Nature Communications' readership.

Reply to Reviewer #1:

The authors propose using the pseudomode formalism to simulate electron transfer in the presence of nucleon vibrational degrees of freedom as a non-Markovian bath and demonstrate the simulation protocol on an IBM QPU. The pseudomode-based digital-analog hybrid approach enables demonstrations to be executed on cloud QPUs without the extensive calibration information required by approaches that rely on residual ZZ interactions. While it is unclear whether this approach is scalable to a quantum utility level, the demonstration of simulating open-electron-transfer dynamics and the approach potential in the NISQ regime are worth publishing. I recommend that this manuscript be published after considering the following comments.

We are grateful to the Reviewer for the positive remarks on our work and their careful reading. The comments are insightful and have made us reconsider the presentation of some key ideas in the main text.

Main Comments:

- 1) **(Rev.1-1)** *As the authors noted, the 1D dissipative model with the pseudomode approximation is well-suited to classical tensor network simulation. It seems unlikely that increasing the chain size as suggested in Fig. 3b would provide a favorable scaling for the quantum simulation over the classical algorithm. It would be helpful if the authors could suggest in the manuscript under what circumstances their approach could potentially provide an advantage over the classical one.*

We thank the Reviewer for raising this important point regarding the prospective quantum advantage of our approach. We have removed the subtitle “A Scalable Approach to Open Quantum Systems” from the title to better reflect the specific scope of our experimental demonstration. We have also added two new paragraphs to the Discussion section that explicitly analyze the algorithmic complexity of our method using $\mathcal{O}(\cdot)$ notation and compare it directly to the classical tensor-network DAMPF method [Phys. Rev. Lett. 123, 100502 (2019)], which we used for our classical simulations on a high-performance computing (HPC) cluster.

We emphasize that the primary challenge in simulating open quantum systems lies not in the complexity of the vibrational environment per se, but in the scaling of the system size; specifically, the number of electronic sites N . While classical methods such as HEOM or T-TEDOPA are highly effective for small systems (e.g., $N=2$) with structured environments [arXiv:2601.02160 (2026)], they face severe limitations when scaling to larger networks. In contrast, our quantum algorithm features constant circuit depth with respect to N , enabling efficient scaling.

We agree that the model depicted in Fig. 7 **b** in the revised manuscript (Fig. 3 **b** in the former version), which shows $N=50$ sites with one oscillator per site $M=1$, will reach quantum advantage in the regime of weak vibronic coupling only for system sizes of at least $N=100$ sites, as classical methods like DAMPF can efficiently compress vibronic states when electronic and nuclear states are weakly entangled, using efficient tensor-networks representations. However, a clear advantage emerges in non-perturbative regimes characterized by strong vibronic coupling ($g \sim J$) and weak damping ($\gamma \rightarrow 0$). In such cases, classical simulations require a large number of oscillator levels (N_b) and high bond dimensions (χ) for convergence, leading to a memory scaling of $\mathcal{O}(N^3 M N_b^2 \chi^2)$ in the single-excitation manifold, a severe bottleneck. In contrast, our quantum algorithm requires only a linear increase in the number of qubits $\mathcal{O}(N M N_b)$, and due to the parallelization of gates on sites, the depth of the Trotter circuit depends only on the vibrational environment $\mathcal{O}(M N_b)$ and remains constant when increasing the system size N . Most importantly, the quantum algorithm is independent of the bond dimension χ , as the quantum processor naturally encodes the full vibronic state in the ansatz defined

by our Trotterized quantum circuits. In contrast, for classical methods the bond dimension χ may grow exponentially when increasing vibronic coupling g at fixed numerical accuracy, leading to an exponential scaling in both memory and time with respect to these parameters. We note that the inclusion of multiple vibrational modes (Fig. 7 c), ultra-strong vibronic coupling or broader spectral features were already demonstrated in prior work by our co-authors for small systems [Phys. Rev. A 108, 062424 (2023)].

Moreover, it is important to emphasize that the $\mathcal{O}(N^3 M N_b^2 \chi^2)$ scaling of DAMPF already assumes the restriction to the single-excitation manifold. Classical simulations of multi-excitation dynamics (e.g., explicit treatment of multiple orbitals per site) would face exponential scaling in the electronic Hilbert space dimension, whereas the quantum algorithm requires only a linear increase in qubit count (e.g., doubling qubits to include both HOMO and LUMO levels per site) with no fundamental change to the simulation protocol.

We gave all these considerations their own place in the Discussion section in the revised manuscript (revised manuscript, lines 670-729):

“To assess how the quantum approach presented here performs relative to state-of-the-art classical techniques, we compare to the tensor-network DAMPF algorithm [59], which scales as $\mathcal{O}(N^3 M N_b^2 \chi^2)$ for a parallelized HPC-implementation of the 1D chain model in the single-excitation manifold as presented here, with N sites and M oscillators per site, N_b energy levels per oscillator and χ as the bond dimension of the matrix product operators, underlining that the main challenge for classical simulations lies in the scaling of the system size N . [...] Although models with more complex vibrational structures are outside the scope of this work, it is also possible to include more oscillators and energy levels in our approach, as shown in Fig. 7 c. Such multi-pseudomode models can be used to fit arbitrary spectral densities $J(\omega)$, including features like thermal environments. If every site is coupled to independent vibrational environments, the number of qubits scales linearly with the number of oscillators M . When using a straight-forward encoding where each additional phonon level is mapped to a single qubit (see Supplementary Methods), the number of required qubits scales as $\mathcal{O}(N M N_b)$. More efficient encodings of oscillators could in principle lead to a logarithmic scaling in N_b . The quantum circuits can be parallelized with respect to N , similar to the Trotter circuit in Fig. 7 a. However, each oscillator qubit that couples to an electronic site adds an additional gate layer, causing the Trotter circuit depth to scale as $\mathcal{O}(M N_b)$. Note that this scaling holds for All-To-All, Square, and Heavy-Hex topologies. However, the latter require several layers of SWAP gates to move the system sites along the oscillator chains. Compared to the scaling of DAMPF (see above), the advantage of our quantum algorithm persists. Crucially, classical simulations quickly become intractable in non-perturbative regimes characterized by strong vibronic coupling ($g_0 \sim J$) and weak damping, which may require exponentially large bond dimensions χ for converged simulations at constant numerical accuracy. The quantum algorithm is independent of bond dimensions χ as the quantum processor naturally encodes the full vibronic state in the qubit-mapped Hamiltonian.”

2) **(Rev.1-2)** *While the authors suggest they present a digital quantum simulation toolbox, their approach appears to be a digital-analog hybrid, as they lack control over the oscillator qubit's dissipation. Beyond this terminology issue, several related aspects require further clarification.*

We thank the Reviewer for highlighting the hybrid nature of our approach. We fully agree that the lack of precise control over dissipation rates in superconducting qubits represents a departure from a

strictly digital quantum simulation. In response, we have revised the abstract and removed the term “digital toolbox” to more accurately reflect the analog-digital hybrid character of our method:

“We present a framework for the digital-analog simulation of open quantum systems with structured vibrational environments. Exploiting the intrinsic dissipation of qubits as a resource to emulate vibrational relaxation, and combined with a model-specific error mitigation scheme to filter out noise sources incompatible with the target open system, our approach enables scalable simulations of non-Markovian dynamics on near-term quantum hardware.”

This phrasing acknowledges that while the time evolution is implemented digitally via Trotterization, the effective damping of oscillator qubits arises from the intrinsic amplitude damping of the hardware, a key analog-like resource.

We emphasize that while arbitrary control over dissipation rates is not currently feasible on superconducting hardware, partial control can be engineered within a practical window defined by the total evolution time T , the time step dt , and the T_1/T_2 values of qubits in the hardware. One of our main goals in this work was to identify the largest vibrational relaxation lifetime that can be simulated on the processor, in accordance to the existing dissipation rates in the current hardware. If the dynamics of a weakly damped model with damping rate γ' can be simulated on the hardware accurately from $t=0$ to $t=T$, then models with stronger damping rates ($\gamma > \gamma'$) can be readily engineered using several approaches. However, the challenge here is precisely the simulation of models with weak damping, giving rise to long-lived vibronic coherence with a strong Non-Markovian character. In this challenging regime, the T_1/T_2 values of the qubits assigned to oscillators should be as large as possible, such that the reduced dynamics of the system can be accurately reproduced within the time interval $(0,T)$.

This issue was partially addressed in the Introduction of the submitted version of the manuscript (lines 105-118). In the revised manuscript, we have refined this paragraph to clarify this point: (Lines 102-128, revised manuscript)

“When simulating the Trotterized dynamics of the closed model on the quantum computer with a time step Δt as shown in Fig. 1, the intrinsic noise of the quantum processor will imprint an effective damping rate Γ_{QC} , which is determined by the T_1 , T_2 times of the involved qubits (see Eq. (18)). Rather than targeting a specific damping rate, this work focuses on maximizing simulation time to identify the longest vibrational relaxation lifetime Γ_{QC} achievable on the quantum processor. The effective damping rate Γ_{QC} can be tuned by adjusting the Trotter step Δt : larger steps reduce dissipation imprinted on the resulting dynamics, while smaller steps increase it, as demonstrated previously [23]. However, this tuning is constrained by two key limits. First, Δt must be sufficiently small to fully resolve the spectral content of the Hamiltonian, determined by its largest possible frequency. Second, excessively small Δt leads to deep quantum circuits, increasing the number of time steps required to reach a given evolution time. In general, target models with stronger damping rates ($\gamma \gg \Gamma_{QC}$) can also be implemented either via delay instructions on environment qubits [32] or alternatively, by coupling the latter to an ancillary qubit with periodic reset operations [61]. However, the simulation of target models with longer vibrational lifetimes ($\gamma \ll \Gamma_{QC}$) requires improved hardware with increased qubit coherence times beyond what is available today.”

In contrast to commercial superconducting platforms, trapped-ion systems offer control over dissipation rates within certain bounds [Sci. Adv. 10, eads8011 (2024), Nature Communications 16, 4042 (2025)]. However, scaling such systems beyond small spin-boson models has proven challenging.

Our work demonstrates that superconducting hardware, despite limited control over dissipation, enables scalable simulations of extended vibronic networks; a capability that remains elusive on other platforms.

3) **(Rev.1-3)** *How do the variations of T_1 and T_2 among the oscillator qubits vary after the simulation? While the authors suggest that the worst qubit dominates the effective damping, as shown in equation 13, don't the variations also play an important role? It seems unlikely that the two cases (a) all oscillator qubits have poor T_1 and T_2 , and (b) only the last oscillator qubit has poor T_1 and T_2 , will give similar results.*

We agree that the statement that *"the worst qubit dominates the effective damping rate"* should not be interpreted as a strict, quantitative rule, but rather as a qualitative guideline. In the following, we justify this perspective and tackle at the end the cases (a) and (b) proposed by the Reviewer with a toy example of $N=3$ sites.

Equation (13) (now Eq. (18) in the revised manuscript) provides a lower bound for the effective vibrational lifetime (Γ_{QC}^{-1}) simulated on the quantum computer. The inequality becomes equal when all qubits have the same damping rate, i.e. when all qubits have the strongest damping rate. In the general case, the inequality reflects the fact that Γ_{QC} is limited by the qubit in the chain with the strongest damping, while the exact value of Γ_{QC} depends on the full distribution of T_1/T_2 times. The qualitative behavior, that the shortest coherence time sets the practical limit, remains robust.

We have clarified this point in the Methods section when introducing Eq. (18), where we emphasize that the minimum T_1/T_2 among the oscillator qubits sets a practical upper bound on the effective damping rate Γ_{QC} . Revised manuscript, lines 902-928:

"For any experiment of Trotterized dynamics on the quantum computer, the correspondence between hardware error rates and the effective damping rate Γ_{QC} that one is able to simulate depends on the hardware execution time $T_{exec}^{Trot.}$ of one Trotter circuit and the particular choice of simulation time step, which we fixed to $\Delta t = 4$ fs for all our experiments on the quantum computer. For our simulations of the ET-model, a reasonable upper bound for the effective damping rate Γ_{QC} is given by

$$\Gamma_{QC} \lesssim \frac{T_{exec}^{Trot.}}{\Delta t} \frac{1}{\min_q (T_1^{(q)}, T_2^{(q)})}$$

where $\min_q (T_1^{(q)}, T_2^{(q)})$ refers to the smallest T_1 or T_2 time among the qubits in the circuit. In the general case, the inequality reflects the fact that Γ_{QC} is limited by the qubit in the chain with the strongest damping. This mirrors kinetic competition in our electron transfer model (depicted in Fig. 1 a): productive vibronic separation and lossy trapping constitute parallel pathways, where the faster channel dominates the quantum yield by outcompeting alternatives. Similarly, among independent decoherence channels acting on different qubits, the fastest decay governs Γ_{QC} , establishing a hardware-imposed ceiling on simulable vibrational lifetimes. Note that Eq. (18) becomes equal for the case of equal damping rates on each qubit, while the exact value of Γ_{QC} depends on the full distribution of T_1, T_2 times. This bound represents a qualitative statement, as it disregards other sources of noise in the quantum computer.

To address specifically the Reviewer's comment *"It seems unlikely that the two cases (a) all oscillator qubits have poor T_1 and T_2 , and (b) only the last oscillator qubit has poor T_1 and T_2 , will give similar*

results.”, we have prepared the following simulation for a simple model with $N=3$ sites. This example is not included in the manuscript, but it provides a direct comparison between the cases (a) and (b) described by the referee, illustrating our argument about the upper bound for Γ_{QC} when oscillator qubits have different T_1 and T_2 values. The population dynamics of sites 0, 1 and 2 at the vibronic resonance are shown in Figure Rev. 1-3 below.

Figure Rev. 1-3: Emulations of a vibronic resonance for $N=3$ sites; the x-axis shows simulation time in fs. **(a)** All oscillators have strong decay rates (blue lines). **(b)** Only the last oscillator is strongly damped (red lines). **(c)** All oscillators are weakly damped. **(d)** Dynamics with no vibronic coupling (yellow lines).

We consider vibrational lifetimes of 200 fs and 100 fs to discern between “good” and “bad” qubits, respectively. For the donor-trap-acceptor model with $N=3$ sites, our figure of merit P_{transfer} is solely determined by the population of the last site (acceptor Site 2). The case (b) “only the last oscillator has poor T_1 and T_2 ” in red has only slightly higher probability of transfer than case (a) “all oscillators have poor T_1 and T_2 ” in blue. More importantly, the probability of transfer of the case (b) lies between the cases (a) “all osc. qubits have poor T_1, T_2 ” and (c) “all osc. qubits have good T_1, T_2 ”, which is consistent with our claim that (a) represents an upper bound when determining Γ_{QC} in the presence of oscillator qubits with different T_1, T_2 values.

4) **(Rev.1-4)** How is g being chosen for the $g>0$ case, and what is its value used in the experiment and simulations?

We thank the Reviewer for pointing out that our label $g > 0$ was very unprecise. We remark that we always chose in all experiments and benchmarks the same value of $g_0 = \omega_0 \sqrt{s}$, where $s=0.05$ is the dimensionless Huang-Rhys factor and $\omega_0 = 1500 \text{ cm}^{-1}$ is the oscillator frequency.

We have updated the notation accordingly from $g > 0$ to $g = g_0$ throughout the manuscript, including the labels in the legend of all Figures. In the revised manuscript, we refined the definition of the model parameter g (Lines 219-223):

“[...] we model this single-peak $J(\omega)$ with a single pseudomode (see Methods), i.e. a single oscillator ($M = 1$) of frequency ω_0 , damping rate γ depending on the width of $J(\omega)$, and vibronic coupling strength $g_0 = \omega_0 \sqrt{s}$, where $s = 0.05$ is the dimensionless Huang-Rhys factor.”

5) **(Rev.1-5)** In Section III B2, the authors state that a target damping rate $\gamma=(1 \text{ ps})^{-1}$ is used to compare the experimental results. However, in Appendix D, the experimental results are compared with simulation results over a γ range of $(12.5 \text{ fs})^{-1}$ to $(500 \text{ fs})^{-1}$, and the best-fitting damping rate is $\sim(100\text{-}200 \text{ fs})^{-1}$. Does it mean that the performance measured in Fig.6 has a weak dependence

on γ for an overall error level of 10%? If so, it seems odd since γ plays an important role in simulating the vibronic resonance.

We thank the Reviewer for this particular question. We realised that the content of Section III B2 overall and the discussion of Fig. 6 (now Fig. 5 in the revised manuscript) in particular, was imprecise and difficult to digest. Given the emphasis that we put on the concept of an “effective vibrational lifetime” Γ_{QC}^{-1} on the quantum computer, it is paramount for us that the reader understands how we extract these parameters from simulations in a clear way. Consequently, we have substantially improved the presentation of this part and added new panels Fig. 5 d-e that were adapted from the plots mentioned by the Reviewer previously appearing in Appendix D. We hope these explanations provide a clear answer to the Reviewer’s question “Does it mean that the performance measured in Fig.6 has a weak dependence on γ for an overall error level of 10%?”.

Revised Fig. 5 (previously Fig. 6): We added the following two panels: **d.** Extraction of Γ_{QC} by comparing the quantum simulation for $N = 7$ sites corresponding to the overall lowest deviation in panels a and b against classical simulations with γ ranging between $(12.5 \text{ fs})^{-1}$ and $(500 \text{ fs})^{-1}$. The dot markers indicate the minimum of each curve. **e.** In comparison, the deviations $\epsilon(T)$ for a quantum simulation with $N = 7$ on a day with more noise, indicated by the overall higher level of deviation.

First, the function $\epsilon(T)$ strongly depends on the value γ used in reference calculations, and this dependence becomes more evident for times $T \geq \gamma^{-1}$. We clarify that the weakly damped model that was mentioned in the caption of the old version of the Figure with a damping rate of $\gamma = (1 \text{ ps})^{-1}$ can be considered as practically undamped ($\gamma=0$), because the difference in the population dynamics between the two cases ($\gamma = (1 \text{ ps})^{-1}$ vs $\gamma=0$) is negligible within the first 300 fs that we used for the plot.

In general, this reference calculation should quantify the hardware's capability to simulate weakly damped dynamics ($1/\gamma \gtrsim 500$ fs), which is the regime that remains most challenging for classical methods. For the sake of clarity we now explicitly compare quantum simulations in panels **a-c** against strictly undamped classical simulations ($\gamma=0$), instead of $\gamma = (1 \text{ ps})^{-1}$.

Second, we have renamed the function $\epsilon(T)$ from 'time-averaged error' to 'time-averaged deviation'. We realized the name 'error' is misleading, as $\epsilon(T)$ merely quantifies a deviation from a particular reference calculation. The same quantum simulation may have a high value of $\epsilon(T)$ when compared against emulations without damping ($\gamma=0$), but a much lower $\epsilon(T)$ value when compared against emulations with some finite damping rate ($\gamma>0$). This is especially clear at later times ($T>\sim 100$ fs) after which deep circuits are heavily influenced by the intrinsic damping of qubits. This scenario is illustrated by the simulation presented Fig. 5 **d**, where $\epsilon(T)$ presents a clear minimum and the value of $\epsilon(T)$ around the minimum stops increasing at later times, i.e. $\epsilon(T + \Delta t) \approx \epsilon(T)$. These are the signatures of $\epsilon(T)$ that we require to extract a meaningful effective lifetime Γ_{QC}^{-1} , indicating the successful reproduction of pseudomode damping. In contrast, a high value of $\epsilon(T)$ could also arise from other noise sources in the quantum computer that are not compatible with the pseudomode formalism. In this case, $\epsilon(T)$ remains large for any finite value of γ used as classical reference and its value keeps increasing at later times $\epsilon(T + \Delta t) > \epsilon(T)$. The experiment in Fig. 5 **e** corresponds to such a case with increased levels of $\epsilon(T)$, where the extraction of a meaningful effective damping rate Γ_{QC}^{-1} is harder as pseudomode damping is overshadowed by a stronger presence of other noise.

In conclusion, comparing the experiments on the quantum processor against emulations with $\gamma=0$ benchmarks the capability of the hardware to simulate weakly damped dynamics ($1/\gamma \gtrsim 500$ fs), a regime of high utility over classical methods, while the comparison against simulations with $\gamma>0$ validates the pseudomode experiment on the quantum computer. For this reason, we believe that it was important to complement Fig. 5 in the main text with the new panels **d** and **e**, adapted from the "best-fitting" analysis with varying γ that was previously only shown in Appendix D.

We reflected these changes in the text (Lines 457-473):

"The deviation function $\epsilon(T)$ strongly depends on the value γ used in reference calculations, and this dependence becomes more prominent for times $T \gtrsim 1/\gamma$. First, we consider strictly undamped ($\gamma = 0$) reference calculations, corresponding to unitary dynamics without dissipation. A high $\epsilon(T)$ relative to $\gamma=0$, for example, does not necessarily indicate poor performance, as this deviation can be significantly reduced when comparing against reference calculations with finite damping ($\gamma > 0$). First, we compare our experimental results on the quantum processor to undamped ($\gamma = 0$, Fig. 5 a-c) reference simulations, providing a clear benchmark that quantifies the hardware's capability to simulate long-lived, weakly damped dynamics ($1/\gamma \gtrsim 500$ fs), a regime that remains challenging for classical simulations [5, 47]. Furthermore, we compare experimental results against damped ($\gamma > 0$, Fig. 5 d-e) reference simulations, which allows to determine the effective damping rate Γ_{QC} imprinted onto the pseudomode ET-model."

Minor comments:

- 6) (Rev.1-6) What is the justification to assume that each site is coupled to an independent vibrational environment? If the frequency ω_0 is associated with a specific set of intramolecular vibration modes, shouldn't the electron be coupled to the same set of modes at different locations in the potential well?

As depicted in the schematic of Fig. 2 a, we consider a model of neighboring fullerene molecules coupled to a single donor, each of them represented by the LUMO level of each monomer (site). We only consider intramolecular, dispersionless vibrations of the high-frequency C=C stretch modes, which can be treated as independent vibrational modes that are locally coupled to each individual molecule. Thus, in a discrete representation of the potential well, the electron is **not** coupled to the same set of modes at different locations, but rather the electron couples to an independent set of vibrational modes at each location. Nonetheless, because all monomers (sites) in the chain are identical, we simply assume that each set of modes have the same frequencies and vibronic coupling strengths to their respective LUMO levels. In other words, every site is locally coupled to a vibrational environment with identical spectral densities, as explicitly stated by Eq. (5), now Eq. (6) in the revised manuscript. We emphasized this point more clearly in Line 211-216:

“We focus on modeling high-frequency, intramolecular vibrations like C=C stretch modes that are intrinsic to organic molecules. Therefore, we assume that every site is coupled to identical vibrational environments that are independent from each other [...]”

The high-frequency stretch modes in our model have very long lifetimes and they are the fundamental origin of the vibronic mechanism of charge separation. We note that dispersive, low-frequency phonons, which we did not consider in this work, lead to non-local, vibronic coupling terms where each mode is simultaneously coupled to all sites at different coupling strengths. Nonetheless, such low-frequency vibrational modes are routinely modeled as uncorrelated environments using standard spectral densities at each site, such as an Ohmic background with an exponential cutoff.

7) (Rev.1-7) Table I suggests that the amplitude damping, pure dephasing, and depolarization of the site qubits need to be fully mitigated. In the experiment, is it correct that only the amplitude damping and depolarization are partially mitigated by the postselection, and the pure dephasing is not mitigated? Also, while randomized compiling and (partial) probabilistic error cancellation are mentioned, are they used in the experiment?

It is important to remark that we do **not** apply randomized compiling nor probabilistic error cancellation (PEC) as part of our error mitigation strategy. PEC requires a large number of experiments to characterize error channels and becomes impractical for the number of qubits that we are employing. Besides, randomized compiling leads to deeper circuits when applying Pauli twirling, such that we would transform coherent, gate errors into incoherent noise with an increased effective depolarising rate. We further stress this technical detail below in our answer Rev.1-10.

Our post-selection is indeed based exclusively on qubit populations and discards shots that were incompatible with conserved quantities. We only sampled qubits measured in the computational basis to retrieve the population dynamics, we do not need to deploy additional measurements to apply error mitigation. Therefore, our error mitigation protocol can correct qubit populations, while coherence lost to pure dephasing cannot be genuinely recovered by these means.

We clarified these points in the Methods section (Lines 950-955):

“We employ two levels of a model-specific error mitigation scheme, that is hardware- and implementation-agnostic and can correct for depolarizing errors and amplitude damping on electronic sites (see Tab. I). It is complemented with readout error mitigation [68] for the results presented in this work.”

And further down in the same section (Lines 979-986):

“When combined with noise characterization and error mitigation techniques like randomized compiling and probabilistic error cancellation, partial control over error rates in the qubits could be achieved [26]. However, we note that since one goal of this work is to maximize vibrational lifetimes and maintain system-bath coherence as long as possible, we did not apply such mitigation techniques that would address coherent gate errors.”

8) (Rev.1-8) p.4: *“The lifetime of vibrational excitations is inversely proportional to the width of the peak in the spectral density $J(\omega)$ (see Fig. 1 b).” It should probably be Fig. 1a instead.*

Indeed! We have corrected this typo.

9) (Rev.1-9) p.5: *“Our quantum circuits simulate the closed model in Fig. 1 a with $\gamma = 0$ using Trotterized quantum evolution on the quantum computer and 104 shots per time step.” The simulations shown in part III use $g = 0$ instead of $\gamma = 0$.*

We thank the Reviewer for pointing out this unclear statement. Our intention was to elaborate that although the target model is open, our quantum circuits by themselves correspond to the Hamiltonian and do not contain information about the damping per se.

The revised version of the sentence, that was further altered due to the swapped order of Methods and Results section, reads (Lines 266-270)

“We perform quantum simulations of this model as depicted in Fig. 1 a by using quantum circuits corresponding to the full Hamiltonian in Eq. (1), employing Trotterized quantum evolution on the quantum computer with 10^4 shots per time step”

10) (Rev.1-10) p.6: *“On the other hand, imperfections in the hardware implementation of single and two-qubit gates introduce depolarising noise that progressively destroys quantum coherence among qubits.” The gate errors effectively reduce to depolarising noise only when randomized compiling is implemented. It is also technically incorrect to include the gate error in equation 12, since the effective depolarising noise is at the level of a randomized compiled Trotter step block rather than the continuous master equation evolution.*

This question complements our answer Rev.1-7 above, where we remark that our error mitigation strategy does not include randomized compiling, which indeed turns *coherent noise* into *incoherent noise* (depolarising noise). This is a common issue with the nomenclature used in quantum computing when referring to “gate errors” or “gate imperfections”.

Coherent noise can be understood as a constant bias or *overrotation* deviating from the desired angle of a parametrizable gate. This type of noise can be considered as static because the overrotation values fluctuate in much slower timescales than the typical execution time of any of our circuits. We stress that we do not apply any mitigation technique addressing coherent noise on purpose, because our goal is precisely to achieve the longest possible coherence time (vibrational lifetime) on the hardware. In that regard, the depolarization rate entering as a prefactor in Eq. (12) (now Eq. (17) in the revised version) should be understood as broadband, stochastic fluctuations (incoherent noise) in the control pulses that lead to depolarising noise during the execution of the gate.

We addressed these points in the text (Lines 864-878):

“We assume that noisy processes in the hardware stem from two major sources. One noise channel contains amplitude damping and pure dephasing, which are continuously affecting all qubits, including idle qubits. Another noise channel is depolarisation from gate noise. On the other hand, imperfections in the hardware implementation of single and two-qubit gates introduce additional

detrimental effects such as depolarising noise that progressively destroys quantum coherence among qubits. Although we acknowledge there are other sources of noise, it is sufficient for our purposes to consider noise channels that can be described by Markovian theory. After mapping the full Hamiltonian Eq. (1) to operators acting on qubits in the processor, we model hardware noise with a master equation within the Lindbladian formalism [...]"

Reply to Reviewer #2:

In this work the authors perform a quantum simulation, using an IBM-Q cloud quantum computer, of electron transfer through a one-dimensional chain, where each site is coupled to local non-Markovian environment (which are included in the simulation via ancilla qubits). Applications to the study of light-harvesting and photo-cells are discussed.

The idea of simulating open-quantum systems on quantum computers is seeing growing interest, both for purposes like the authors explain in this work, and to more abstract problems, like finding many-body ground-states via dissipative state engineering. I found this article to be well written, and clearly explained, with a wealth of detail on the steps they made to implement their protocol that will surely be useful for future works.

The novelty in this work is largely in that detail, and in the simple but useful steps the authors make in improving and analyzing their results (like the error mitigation steps based on discarding shots that don't conform to physical expectations of their model, and using the intrinsic dissipation of the qubits themselves to get the correct ancilla environmental noise). On the positive side, their study implies larger scale simulations, at least of models similar to the ones they construct are feasible in the near future, and could be actually useful. On the negative side, the framework they build up is made of largely known concepts, so I felt, at least from the theoretical point of view, the impact might not be so strong. In addition, moving away from many of the model assumptions they make seems difficult, and no analysis of the impact of doing so is provided.

We thank the Reviewer for their assessment, and take their concern regarding a missing analysis of more general model assumptions very serious. As similar remarks about the generability of our methodology have been made by the other Reviewers, we have removed the subtitle “A Scalable Approach to Open Quantum Systems” from the title. Furthermore, we created a Discussion section where we investigate the computational complexity when simulating more complex models, for which we refer to Rev. 1-1.

Comments and Questions

I have a few questions and comments that I hope the authors can consider which may help improve the work on these points:

11) **(Rev.2-1)** *Am I correct in understanding an important step is to make sure that the "worst" qubits are chosen for the pseudomode ancillas? And that, in the end, the 'simulation parameters' are essentially given by the hardware? There was some discussion I guess about expanding the run time of the circuit, to effectively increase that dissipation if needed, but this was not done in practice in any of the examples, is that correct?*

We thank the Reviewer for this particular question. A similar remark was pointed out by the first referee regarding the lack of precise control over the dissipation rates of qubits. This compelled us to improve our explanations in the manuscript, and we refer to our answers Rev.1-2 and Rev.1-3 above to complement this answer.

We have argued that our main goal in the present work was to identify the largest vibrational relaxation lifetime that can be simulated on the quantum computer. For this reason, we always choose the best qubits available on the processor for both system and oscillator qubits, while giving preference to the former. We have clarified this point in the introduction section of the revised manuscript, lines 102-124:

“When simulating the Trotterized dynamics of the closed model on the quantum computer with a time step Δt as shown in Fig. 1, the intrinsic noise of the quantum processor will imprint an effective damping rate Γ_{QC} , which is determined by the T_1 , T_2 times of the involved qubits (see Eq. (18)). Rather than targeting a specific damping rate, this work focuses on maximizing simulation time to identify the longest vibrational relaxation lifetime Γ_{QC} achievable on the quantum processor. The effective damping rate Γ_{QC} can be tuned by adjusting the Trotter step Δt : larger steps reduce dissipation imprinted on the resulting dynamics, while smaller steps increase it, as demonstrated previously [23]. However, this tuning is constrained by two key limits. First, Δt must be sufficiently small to fully resolve the spectral content of the Hamiltonian, determined by its largest possible frequency. Second, excessively small Δt leads to deep quantum circuits, increasing the number of time steps required to reach a given evolution time. In general, target models with stronger damping rates ($\gamma \gg \Gamma_{QC}$) can also be implemented either via delay instructions on environment qubits [32] or alternatively, by coupling the latter to an ancillary qubit with periodic reset operations [61].”

The dissipation rates are determined by the Trotter step Δt of the simulation and the intrinsic qubit dissipation. As demonstrated by co-authors in a previous work [Phys. Rev. A 108, 062424 (2023)], the easiest way to manipulate damping rates in quantum simulations is by changing the Trotter time step. Choosing a larger (smaller) Δt decreases (increases) the effective dissipation rate in simulations. There are limitations in the range of values that Δt can take: on the one hand, Δt cannot be larger than half the period of the largest frequency in the Hamiltonian spectrum; on the other hand, small values of Δt lead to quantum circuits that are too deep, requiring more time steps to reach the same evolution time than simulations with a coarser time step. In the simulations we presented, we chose a fixed time step that balances Trotter error with circuit depth in order to increase the total time of the simulation as much as possible. Lines 832-839:

“For all our simulations on the quantum computer, we evolve an initial excitation in the donor site with a fixed time step $\Delta t = 4$ fs. This value represents a good compromise between total circuit depth and Trotter error and is motivated by the fact that the highest frequency in our Hamiltonian (Eq. (1)) is around $(11 \text{ fs})^{-1}$, stemming from the large Coulomb binding energy $V = 2420 \text{ cm}^{-1}$ of bound electron-hole pairs in organic molecules.”

12) (Rev.2-2) I was wondering, is it given (1) is it possible to change the ancilla dissipation, without changing the "system" qubit dissipation as well? Or would one have to employ more sophisticated steps, like yet more ancillas + readout?

One would need indeed to deploy more ancillas + readout, to avoid the influence of engineered dissipation on "system" qubits as much as possible, which are assumed to be noiseless. This is indeed a non-trivial problem that we do not address in the manuscript. Generally, there are many options to influence oscillator dissipation rates, using for example idle gates or additional ancillas with mid-circuit measurements. However, note that all these methods can only increase dissipation, while in practice, the amount of hardware noise on current devices is already too strong for any of these methods to be useful at all.

Besides, classical models perform very well for strongly damped vibrational environments, as the dynamics are generally well described by Markovian theory. For this reason, our focus are narrow peaks in spectral densities, as they often represent the greatest challenge in classical simulations of open quantum systems.

13) **(Rev.2-3)** A minor comment, below equation 8 the authors state " We fix the temperature to zero, where the bath correlation function $C(t)$ and the spectral density $J(\omega)$ form a Fourier pair." While true for the very narrow Lorentzians ($\gamma \ll \omega_0$) the authors employ here, in general this is not the full story; as they mention at the end of the page, $J(\omega)$ is only defined for positive frequencies, so I believe extending the integral to negative frequencies requires taking this account.

It is always possible to express the reservoir correlation function either by its symmetric or antisymmetric part $C^{(\pm)}(t) = C(t) \pm C^*(t)$. The half-sided Fourier integral defining the bath correlation function $C(t)$ in Eq. (8) should be kept at positive frequencies, because it already takes into account both contributions.

The motivation to employ an antisymmetrized Lorentzian spectral density is to avoid the non-vanishing coupling at zero energy that is present when using a simple Lorentzian function [e.g: Phys. Rev. A 101, 052108 (2020), Sci. Adv. 11, eady6751 (2025)]. We do not employ an antisymmetrized Lorentzian because artifacts stemming from non-vanishing coupling at $\omega=0$ become negligible for narrow Lorentzians ($\gamma \ll \omega_0$) centered at high frequencies $\omega_0 \gg 0$, which is our current focus. We provide a short justification in the following and we refer to Chapter 3.6 from Rev. 92 [V. May and O. Kühn, *Charge and Energy Transfer Dynamics in Molecular Systems*] for further details.

In the theory of open quantum systems, the spectral function $C(\omega)$ is defined as the Fourier transform of the bath correlation function $C(t)$

$$C(\omega) = \int_{-\infty}^{\infty} dt C(t) e^{i\omega t},$$

with its symmetric and antisymmetric parts defined as $C^{(\pm)}(\omega) = C(\omega) \pm C(-\omega)$. The values of the spectral function for positive and negative frequencies are thermodynamically related via a thermal factor $C(\omega) = e^{\beta\omega} C(-\omega)$, with respect to the thermal equilibrium of the reservoir at inverse temperature β . This allows us to express the spectral function in terms of its antisymmetric part only:

$$C(\omega) = (1 + n(\omega)) C^{(-)}(\omega),$$

where $n(\omega)$ is the Bose-Einstein distribution. From this relationship, it also follows that $C^{(+)}(\omega) = \coth \frac{\beta\omega}{2} C^{(-)}(\omega)$. We now write $C(t)$ as the the inverse Fourier transform of $C(\omega)$:

$$C(t) = \frac{1}{2\pi} \int_{-\infty}^{\infty} d\omega e^{-i\omega t} (1 + n(\omega)) C^{(-)}(\omega),$$

which we can express in terms of the half-sided Fourier integral:

$$C(t) = \frac{1}{2\pi} \int_0^{\infty} d\omega \left(e^{-i\omega t} (1 + n(\omega)) C^{(-)}(\omega) + e^{+i\omega t} n(\omega) C^{(-)}(\omega) \right).$$

For Hamiltonians with linear vibronic coupling (LVC) to a discrete set of harmonic oscillators with coupling strength g_ξ , it can be shown that the spectral function takes the following form:

$$C(\omega) = \sum_{\xi} g_{\xi} \left((1 + n(\omega_{\xi})) \delta(\omega - \omega_{\xi}) + n(\omega_{\xi}) \delta(\omega + \omega_{\xi}) \right).$$

To separate the contributions from vibronic coupling and the density of states (DOS) at every frequency ω_ξ , it is convenient to rewrite the spectral function in terms of the spectral density of the environment $\mathcal{J}(\omega) = \sum_\xi g_\xi^2 \delta(\omega - \omega_\xi)$, which is real and positive by definition:

$$C(\omega) = 2\pi \left(1 + n(\omega_\xi)\right) (\mathcal{J}(\omega) - \mathcal{J}(-\omega)).$$

Given that $C(\omega) = (1 + n(\omega))C^{(-)}(\omega)$, we also find that

$$C^{(-)}(\omega) = 2\pi (\mathcal{J}(\omega) - \mathcal{J}(-\omega)).$$

Finally, we arrive at the bath correlation function expressed as an integral over positive frequencies:

$$C(t) = \int_0^\infty d\omega \mathcal{J}(\omega) \left(\coth \frac{\beta\omega}{2} \cos \omega t - i \sin \omega t \right).$$

Spectral densities are often categorized according to the power of ω best approximating their behavior near the origin, where they are always zero.

14) **(Rev.2-4)** *Following on from this, the authors do not particularly explore issues arising from broad environments. They mention finite temperatures, and multiple Lorentzians, so in some sense the components one would need are there, but a bit more detail, or a recipe for general cases, would be useful. This problem has been explored, using pseudomodes, in dissipative state engineering applications, and ideas from those explorations seem useful here too (like using coupled pseudomodes, unphysical pseudomodes, etc), see e.g., <https://doi.org/10.1063/5.0283315> and <https://doi.org/10.1103/PhysRevResearch.6.043229>. It may also be good to cite the original pseudomode framework paper from Garraway and others B. M. Garraway, Phys. Rev. A 55, 2290 (1997), *ibid.* A 55, 4636 (1997), A. Imamoglu, Phys. Rev. A 50, 3650 .*

We now explicitly talk about the possibility to simulate more general environments using pseudomodes, and have added citations to the pioneering works of Garraway and Imamoglu on pseudomodes, which are now cited in the introduction, lines 78-85:

“Our approach uses the pseudomode formalism to simulate the dynamics of the open vibronic system [51–57], depicted in Fig. 1 a. Using auxiliary harmonic oscillators (pseudomodes) that are locally damped by their respective Markovian baths, one is able to engineer arbitrary spectral densities to simulate the non-Markovian dynamics of the reduced system of interest (the electronic network).”

Furthermore, we now emphasize the usage of the pseudomode formalism in the Discussion section, in particular for the simulation of broad environments (lines 699-704):

“Although models with more complex vibrational structures are outside the scope of this work, it is also possible to include more oscillators and energy levels in our approach, as shown in Fig. 7 c. Such multi-pseudomode models can be used to fit arbitrary spectral densities $\mathcal{J}(\omega)$, including features like thermal environments.”

However, we emphasize that complex environments were beyond the scope of this work, as we focused on scaling up the system size; in Rev. 1-1 we argue that the primary challenge in simulating extended open quantum systems with local vibrational environments lies in the system size rather than the complexity of the vibrational environment per se. Furthermore, we note that cases with structured environments, ultra-strong vibronic coupling or broader spectral features were already studied with emulations of quantum computers in prior work by our co-authors for small system sizes [Phys. Rev. A 108, 062424 (2023)].

The influence of a thermal background, given for example by an Ohmic spectral density with some exponential cutoff, can be readily implemented within our quantum algorithm using coupled pseudomodes [Phys. Rev. A 101, 052108 (2020)]. Nonetheless, the engineering of accurate spectral densities $\mathcal{J}(\omega)$ at low frequencies ($\omega \rightarrow 0$) in superconducting qubits for finite temperatures requires further attention and will be the focus of subsequent studies.

At last, we remark that the vibronic mechanism of charge separation, which was the focus of this work, fundamentally originates from the long-lived high-frequency vibrational modes. The impact of a low frequency, thermal background to $\mathcal{J}(\omega)$ can be described as a source of dephasing noise, leading to a broadening of the discrete electronic resonances in P_{transfer} as a function of the driving force. In contrast, the vibronic resonances induced by high-frequency noise are quite robust to the presence of this thermal background. We refer to Fig. 2 in [Commun. Phys. 6:65 (2023)], which clearly illustrates the impact of low-frequency vibrations on the probability of transfer.

15) **(Rev.2-5)** *The authors mention potential scaling upto 100 qubits given better gate fidelities in the near future; however, given the single excitation subspace their particular example employs, and the MPS approaches available to included the dissipative environment, could this case be said to eventually give rise to a model which is out of reach classically? A little bit more discussion on this would be useful.*

As similar remarks about the potential quantum advantage of our methodology over classical techniques have been made by the other Reviewers, we refer to Rev. 1-1 to complement this answer.

Addressing the single excitation subspace mentioned by the Reviewer, it is important to note that the simulation of a single excitation is already challenging for classical techniques in the presence of highly structured environments; the scaling discussion in Rev. 1-1 shows regimes within the single excitation subspace that yield a quantum advantage of our method. While our usage of the single excitation subspace aligned well with our error mitigation scheme, our approach extends directly to multiple excitations for models without double-occupancy on sites, and to models with double-occupancy by employing two qubits per system site. As multi-excitation models would require slightly more complex gates to maintain parity of the fermionic wavefunction, we expect such simulations to become feasible with forthcoming hardware improvements.

At last, to put the scaling of models with up to 100 qubits into perspective, a classically challenging area is for example $N \approx 10$ sites and $M \gg 1$ oscillators per site. As a benchmark example of an extended system with complex vibrational environments, we can consider the Fenna-Matthews-Olson (FMO) complex in quantum biology, which has served as a long-standing challenge in the simulation of non-Markovian dynamics. While well-established classical methods like HEOM or T-TEDOPA may require several days for simulations of a small FMO dimer ($N=2$, $M=31$), latest state-of-the-art tensor-network simulations of the FMO complex employed $N=7$ sites and $M=31$ oscillators per site [Sci. Adv.11, eady6751(2025)]. We estimate a quantum simulation of the dimer model to require around 100 qubits, and a simulation of the full FMO complex to require 300-400 qubits.

16) **(Rev.2-6)** *Many of the technical implementation details are focused on the IBM-Q device they employ. But, the logic and approach seems generally useful for other devices. The text however often switches back and forth between generality and specificity. Given how soon particular devices become obsolete, and the pace of this technology, a small adjustment the presentation may make the results here more transparent and future-proof (e.g., presenting a recipe for analyzing and fixing errors in these types of devices, with the IBM results as a particular example). I think the components are there, it is just a small change in logical flow of the text.*

We appreciate the insight and have adapted the manuscript accordingly. Whenever applicable, we emphasized the generality of the presented framework and separated from the application to IBM, in particular in the last two paragraphs of the introduction, and in Methods for the algorithm itself and for error mitigation.

We note that while our algorithm itself is indeed hardware-agnostic, different hardware shows convenience for different types of models: Superconducting platforms show a good scaling for extended systems sizes, while ion traps can capture detailed bath structures well [Sci. Adv. 10, eads8011 (2024), Nature Communications 16, 4042 (2025)]. Moreover, the hardware of IBM is very suitable to simulate open quantum systems using pseudomodes due to the large presence of amplitude damping noise. Generally, the type of hardware-noise has to be reflected in the Lindbladian of the target open system.

We have adapted the methods section accordingly for the general quantum algorithm (Lines 881-887):

“The core idea is to match the Lindbladian of the target open quantum system in Eq (7) to the noise operators L_n^α . While this approach works generally on any type of quantum hardware, target models with pseudomodes are well suited to superconducting hardware due to the large availability of amplitude damping.”

... and for the error mitigation (Lines 950-955):

“We employ two levels of a model-specific error mitigation scheme, that is hardware- and implementation-agnostic and can correct for depolarizing errors and amplitude damping on electronic sites (see Tab. I).”

Reply to Reviewer #3:

This article proposes protocols for the quantum simulation of open quantum systems and it presents specific results for non-equilibrium electron-transfer dynamics.

The general idea for the quantum simulation protocol is based on the pseudomode model, a theory used for the non-perturbative analysis of Gaussian open quantum systems. It consists in replacing a Gaussian environment with a discrete set of lossy harmonic modes whose parameters are chosen in order to reproduce, or at least approximate, the environmental correlation function of the original bath. The authors consider different possible layouts in which the qubits representing the system are coupled to the qubits representing the environmental pseudomodes. Even within the limitation imposed by the interaction pattern in a quantum chip, the authors envisage the possibility to have multiple pseudo-degrees of freedom interacting with each of the system qubits using swap gates. Interesting extensions of this idea could involve actual harmonic modes to represent the pseudomodes, for example using trapped ions.

As a specific example, the authors consider the simulation of a model for exciton dissociation at an interface made by electron donor-acceptor materials. This model consists in a tight-binding in which the first site corresponds to the donor and the remaining sites to the acceptor. The on-site biases in the acceptor encode the effect of the Coulomb attraction between the electron and the hole in the donor, whose bias, or the detuning (driving force) with respect to the first site of the acceptor chain, is here considered as a free parameter.

The tight-binding model is further complemented by the presence of bosonic environments coupled to each of the sites to describe molecular vibrations. These environments are modeled using the pseudomode model. In particular, each environment is approximated using a single pseudomode, and further represented with a single qubit.

As shown in [76], which includes one of the authors, the charge transfer away from the interface can involve two different physical mechanisms, depending on the intensity of the driving force. For weak driving forces, the energy of the donor can be approximately on resonance with some of the extended eigenstates of the acceptor. The resulting hybridization can then lead to charge propagation away from the interface. While stronger driving forces are going to be detuned with respect to the eigenenergies of the acceptor, they can still be on resonance with "vibrationally hot" (i.e., including a vibrational excitation) delocalized states, leading to a "vibration-assisted exciton dissociation" [76]. Importantly, the phonons involved in this mechanism are characterized by a high frequency spectrum. While the description of a generic Gaussian environment might require several pseudomodes, in this case, the high-frequency characteristic of the spectrum makes it suitable for a single mode description. In fact, since each pseudomode is characterized by a Lorentzian spectral density, its peak at high-frequency mitigates nonphysicalities at zero and negative energies. I also note that, as shown in [76], the performance of the environment in assisting the propagation away from the interface degrades at intense damping, thereby further justifying the single-mode approximation in the case of a narrow spectral density.

In this context, this article reproduces some of the results of the classical numerical simulation reported in [76] by performing a quantum simulation using up to 20 qubits. The gates are defined through Trotterization of the dynamics and their error is modeled in terms of an effective Lindblad equation acting on both system and pseudomodes qubits. In this way, the noise affecting the environmental-qubits is also used to model the broadening of the pseudomode spectrum. To further improve the visibility of the data, the authors also considered a mitigation scheme in which the raw measurements

results are post-processed to discard instances in which particles are not conserved in the system and in which the number of vibrational excitations exceeds a certain threshold. The authors provide the results of different simulations, mostly focusing on the analysis of the time-averaged electron-transfer probability which is a function of the overall simulation time and the driving force. The results of the simulation are compatible with the main physical results mentioned above and obtained in [76]. Furthermore, the experimental data are compared to both classical emulation of the quantum circuit and a classical simulation following the numerical techniques introduced in [76].

Overall, I personally find the main topic of this article, i.e. the simulation of structured open quantum systems on a quantum computer, very interesting. The possibility to simulate quantum effects originating from the mediation of a quantum environment could constitute a very relevant and potentially impactful field of research for quantum technology. More specifically, the modeling of the dynamics in these exciton systems can be relevant to improve our main theoretical understanding on the mechanisms underlying charge propagation for potential technological applications in the development of solar cells.

The article is also, in my opinion, very well written. It describes the model, the simulation, and the results extremely clearly. The overall modeling of the circuit errors, including the qubit mapping and the comparison with emulation and simulation also witnesses a precise and methodical analysis whose results I have no reasons to doubt.

I would also like to mention that, in my opinion, the specific model for electronic transfer constitutes a very clever choice. In fact, on one hand, some physical effects of the model rely on system-bath hybridization to allow transitions to extended states, thereby justifying the focus on a quantum simulation. At the same time, the quantum environments considered here, can also be modeled using the simplest possible pseudomode model, i.e., one where each site is coupled to a single pseudomode. In other words, the authors considered an environment which has non-trivial effects on the system while also being simple to describe with pseudomodes, thereby allowing a quantum simulation within the current technological limitations.

While such a precise choice is evidence of the authors' knowledge, it also hints towards a rather specificity of these results, which is in a bit of contrast with respect to the overall generality of the authors' narrative. More precisely, the authors mention that their goal is to "present a toolbox for digital simulation of large open quantum systems" and that this general approach is validated by studying the electron-transfer dynamics as a specific model. However, the choice of focusing on this specific model and dynamics struck me as very clever exactly because it tightly includes all the assumptions required for the simulation to work, so that it might be not so easy to go beyond it.

In other words, rather than presenting a toolbox with a specific application, I feel this article presents a very specific application with some further additional interesting ideas to move forward.

For example, the possibility to model each environment with a single pseudomode (further represented by a single qubit) is likely not going to hold in more general situations. In fact, even here, the parameters are such that the broadening of each mode corresponds to a regime which is close to the one defined as strongly damped in [76]. In this case, I imagine the single mode assumption might start to not be valid anymore. At the same time, the approximation of the harmonic Hilbert space in terms of a qubit is further justified by imposing a threshold on the pseudomode total population to 1, which might not be possible to impose in general situations. It is not really clear to me how to move beyond this without including new ideas.

It is important for me to mention that this is not a criticism with respect to the actual content of this article but, rather, just about the generality of the narrative. In fact, I do understand that, for example, the broadening of the modes is determined by the overall errors building in the simulation which is definitely going to be improved in the future generation of chips. I also understand that the authors do present interesting ideas in order to simulate multiple pseudomodes for each environment using swap gates or using "native bosonic elements" in other platforms.

However, I have to note that, overall, the results of this quantum simulation are really a subset of those presented by the classical techniques in [76], which appear to me much more efficient. At the same time, as mentioned, the very possibility to even perform this simulation relies on choosing the model and its regime extremely carefully. While I do appreciate the ideas for further improvement, it seems to me that they will be very challenging to implement and, in any case, they go beyond the content presented here.

To summarize my point of view, I would like to simply take into account the title: "Simulating Electron Transfer on Noisy Quantum Computers: A Scalable Approach to Open Quantum System". In my opinion, the content would be much better represented by the first half, i.e., "Simulating Electron Transfer on a Noisy Quantum Computer." In other words, I have to highlight the clever specificity, rather than a broad generality, of the content presented. I further have to comment on the fact that, despite this specificity, the quantum nature of the simulation in this model is not, as far as I can see, close to the efficiency of the classical algorithms it was compared to.

At the same time, I personally found the content of this article very interesting, technically solid, and very well presented. I specifically appreciated the clever choice of this specific application alongside the careful analysis and interpretation of errors. The ideas for further extensions are, in my opinion, also very interesting. As a consequence, despite noting a lack of generality, I also believe this article will definitely be a source of inspiration for researchers in this field.

In summary, while I personally much appreciated the quality of this work, the main issue that prevents me from suggesting publication in Nature Communications in the present form is that its results are very specific, thereby not easily generalized into an actual toolbox for the broad community to use. Furthermore, within the specific domain considered here, the results of the quantum simulation are still quite far from the classical techniques which inspired it. As a consequence, I would like the authors to consider revising the generality of the overall narrative. Alongside this, I think it would be necessary to further add some estimates about the possible performance of the quantum simulation in more general regimes, such as to simulate an environment requiring several pseudomodes. One option in this direction could be presenting some estimation for the scalability of error mitigation when used to model several pseudomodes per qubit using the swap gates mentioned in the manuscript. Another option could be to estimate the performances for the simulation of actual harmonic modes, i.e., without the two-level approximation.

In my opinion, without such additional considerations, these results do not show evidence of the scalability.

We thank the Reviewer for their very thorough reading of our work, for their positive remarks, and their criticism that led us to reconsider the narrative with respect to the scalability. To better reflect the specific scope of our experimental demonstration, as suggested by the Reviewer, we have removed the subtitle "A Scalable Approach to Open Quantum Systems" from the title. Moreover, we separated the general framework and experiment more clearly, in order to clarify that the

experiments implement a single oscillator per site while the framework is more generic; for example at the end of the introduction (Lines 131-137):

“In this work, we present a protocol to simulate non-equilibrium dynamics of vibronic networks, and validate the approach by simulating a microscopic model of electron transfer on the superconducting quantum computer ibm aachen. The model consists of an electronic chain of donor and acceptor sites, each of which are coupled to a local vibrational mode.”

As similar remarks about the generality of our methodology and potential quantum advantage over classical techniques have been made by the other Reviewers, we refer to Rev. 1-1 for a full discussion on this issue.

Below, I present a few specific considerations.

17) **(Rev.3-1)** *As mentioned, I really enjoyed reading about the idea for a simulation of several pseudomodes for each environment using the swap gates. However, it is not clear to me how the authors intend to further improve the simulation to actually recover Gaussianity. For example, even in the single pseudomode case considered here, could the authors present some ideas about a strategy to simulate higher excited states of the harmonic Hilbert space?*

We agree that Gaussianity in pseudomodes requires to simulate them with more energy levels, i.e. Fock states. We give a general qubit mapping including an arbitrary amount of Fock states of harmonic oscillators in the Supplementary Methods (Equations 1-4), which we revised to show most general case with an arbitrary number of oscillators per site. More oscillator levels can be included at a linear scaling in circuit depth and qubit number (see also Rev. 1-1), which we argue in the revised Discussion (Lines 699-720):

“Although models with more complex vibrational structures are outside the scope of this work, it is also possible to include more oscillators and energy levels in our approach, as shown in Fig. 7c. Such multi-pseudomode models can be used to fit arbitrary spectral densities $J(\omega)$, including features like thermal environments. If every site is coupled to independent vibrational environments, the number of qubits scales linearly with the number of oscillators M . When using a straight-forward encoding where each additional phonon level is mapped to a single qubit (see Supplementary Methods), the number of required qubits scales as $\mathcal{O}(NMN_b)$. More efficient encodings of oscillators could in principle lead to a logarithmic scaling in Nb . The quantum circuits can be parallelized with respect to N , similar to the Trotter circuit in Fig. 7a. However, each oscillator qubit that couples to an electronic site adds an additional gate layer, causing the Trotter circuit depth to scale as $\mathcal{O}(MN_b)$. Note that this scaling holds for All-To-All, Square, and Heavy-Hex topologies. However, the latter require several layers of SWAP gates to move the system sites along the oscillator chains.”

We point out that the quantum simulation of oscillators with more levels, using the mentioned oscillator encoding, has already been studied with emulations of quantum computers by our co-authors [Phys. Rev. A 108, 062424 (2023)]. The goal of our manuscript was to explore increasing system sizes and the feasibility on real quantum hardware. Applying our work to models using oscillator with more levels will be subject of future work.

18) **(Rev.3-2)** *Some of the heatmap plots are used to represent the occupation probability at specific sites. Despite the discrete nature of the "position-label", the plots appear to be continuous. I understand that, in [76], this was explained as "the continuous nature of the contour plot emerges when displaying curves of equal populations that are extracted from the time-series*

data."However, I am not sure I really understand this point and I would then like to ask the authors whether they could add a comment explaining this feature in more detail.

We thank the Reviewer for pointing out this lack of explanation, and their reference to an analogous explanation in Ref [76]. When choosing a way of plotting our results, we wanted to ensure that the population dynamics can be deciphered as clearly as possible. Treating the "position-axis" as a discrete variable would result in 5 block columns, where small fluctuations or fast population jumps are not that easy to recognize. Instead, we chose a contour plot, which displays "altitude lines" of areas with equal population. To do so, it linearly interpolates between positions and between timepoints.

We added a comment to the caption of Figure 3:

"To enhance visibility for the population dynamics over few sites and many time steps, the contour plot shows lines of equal population, using a linear interpolation of the position coordinate."

19) **(Rev.3-3)** *As I am not an expert on the physics behind the charge transport, I would like to ask the authors whether the presence of an environment for each of the sites is necessary to justify the propagation away from the interface. For example, would the same effect still be present in case only the first site of the acceptor was coupled to its environment. Is it necessary to couple all the sites to achieve the same transport results?*

The site-environment coupling required to facilitate transport strongly depends on the eigenstate structure of the electronic subsystem. As we elaborate in Supplementary Figure 1 and the Supplementary Notes, ultrafast vibronic charge transport takes place if the initial state is a superposition of two electronic eigenstates, where one is localized in the Donor and the other in the Acceptor, and separated by one oscillator energy. Then, transfer is facilitated by the oscillators coupling to those sites where the relevant eigenstates are localized.

We show an example for N=5 sites in Figure Rev.3-3 below, where the vibronic resonance is made up by a superposition of two states, one localized only in the Donor site, and one mostly localized at Sites 2 and 4 in the acceptor, but also small contributions on Sites 1 and 3.

Extract from Supplementary Figure 1: Eigenstate analysis of the electronic term of the Hamiltonian of the donor-acceptor chain for N=5 sites at the vibronic resonance. The height of the bars indicates

the absolute value of the overlap of each eigenstate with local excitations $|i\rangle$ at each site (from $i=0$ to $i=4$).

We observe in Figure Rev.3-3 below that transfer is facilitated by oscillators according to the amount of localization on each site: When removing the oscillators of Sites 1 and 3, the population transferred into the Acceptor only shrinks by a small amount, however when removing the coupling to Sites 2 and 4, the Acceptor population shrinks significantly. Note that some transfer persists in all cases due to the oscillator coupled to the Donor site.

Figure Rev.3-3: Emulation of the vibronic resonance for $N=5$ sites for an undamped bath $\gamma=0$; the x-axis shows simulation time in fs. Blue: All sites couple to their corresponding oscillator, this case is the vibronic resonance simulated in Figs. 3 and 4 in the revised manuscript. Red: The oscillators of sites 2 and 4, which for this resonance are contributing the most to vibronic transport, are not coupled to the system. Green: For comparison, the oscillators of sites 1 and 3 are not coupled.

In conclusion, we have highlighted that when simulating a particular transfer resonance, the contribution of each oscillator to transport depends on the eigenstate landscape, and there are configurations where not all oscillators contribute equally. However, the goal of studying transport is usually to study different regimes, such as different resonances or driving forces, which alters the eigenstate landscape in unforeseeable ways. Therefore, it is usually very important to include the vibrational environment at every site.

List of changes to the manuscript:

In order to align our manuscript to the formatting requirements of Nature Communications, we had to reorder the sections, resulting in a larger number of minor changes that were required to maintain consistency and a clear line of thought. Here we only list the major changes, for a full overview we refer to the attached pdf version with color-highlighted changes.

- We removed the subtitle 'A scalable approach to open quantum systems' from the title.
- We moved the former Fig. 3 down, such that it is now Fig. 7. The former Figs. 4, 5, 6 and 7 are now Figs. 3, 4, 5 and 6.
- We reformulated the Abstract slightly to comply with the limit of 150 words.
- We added a Discussion section that considers the various aspects of scalability, that were raised by the Reviewers.
- We integrated the data shown in the appendix in Fig. 14 into Fig. 5de, and removed Fig. 14 from the manuscript. We altered the simulation shown in Fig. 1c to match it to Fig. 5de.
- In order to comply with the formatting requirements, we divided up the Methods section: The former subsection A is now called 'Electron Transfer Model' and located at the beginning of Results; the former subsection B is moved into the new Methods section at the bottom of the manuscript; for the former subsection C, the introduction now located at beginning of Results section 'Probing transfer resonances', and the former Fig. 3 (now Fig. 7) together with the corresponding paragraph and the comments on algorithmic scaling are now located in the Discussion section. The remaining subsubsections were moved to the new Methods at the bottom of the manuscript, while merging the former subsubsections 'Trotterized Dynamics' and 'Quantum Circuits and Qubit Encoding' into the section 'Quantum Algorithm'. We removed the headers of Results and Methods that previously outlined the subsection structure.
- To comply with the formatting requirements, we removed the former Conclusion section by discarding all those parts that just gave a summary of the manuscript, and merging the remaining statements into the newly created Discussion section.
- We included statements for Data and Code Availability, Author Contributions and Competing Interests.
- We transformed the former Appendix into the Supplementary Information, with Appendix C as Supplementary Methods and the other sections as Supplementary Notes.
- In the Supplementary Methods, we adapted the Qubit Mapping in Eqs. (1-4) to include the more general case with M_n oscillators per site, the previous version showed the special case $M_n=1$.
- Bibliography:
 - Removed citations: According to the required formatting, we have reduced the number of references to 70 by removing the references 1,2,5,8-11,19-20,22,24-27,29-30,34-36,38-41,43-47,58,74,76-77, mostly concerning electrochemistry, from the introduction.
 - Added citations: The pioneering works on pseudomodes from B. M. Garraway and A. Imamoglu have been added (respectively Ref. [51] and [52] in revised manuscript).

Reply to Reviewer #1:

The authors addressed most of my comments made in my previous report in the revision. Nonetheless, the potential advantage of the proposed approach is unclear to me, as explained below. While I maintain my position that the demonstration and the potential of this approach in the NISQ regime make this manuscript worthy of publication, a discussion of generalizing this approach beyond 1D simulations would make it even more appealing. This manuscript will be suitable for publication in Nature Communications after the suggested change.

In the revision, the authors moderate their stance on their approach being a scalable approach for open quantum system simulation, and focus their discussion on the specific 1D example with an additional complexity analysis. While the complexity analysis is helpful for the readers to understand the potential of the approach, it is important to point out the following three points.

We thank the reviewer for their beneficial assessment, and for their further remarks which again helped to strengthen the manuscript.

Main Comments:

1) **(Rev.1-a)** *For 1D systems, the advantage of the quantum simulation over the classical tensor-network simulation is a polynomial speedup in the system size N and unclear speedup related to the bond dimension of the tensor-network simulation. In my opinion, the $O(N^3) \rightarrow O(N)$ speedup is very unlikely to deliver any quantum utility in the NISQ and early fault-tolerant regimes, given the huge overhead required by QPUs. In the fully fault-tolerant regime, it is unclear, but the digital-analog hybrid approach is not targeted for that regime, nevertheless.*

We agree with the reviewer that it is unclear how much utility the $O(N^3) \rightarrow O(N)$ speedup will deliver in practice on NISQ devices.

For completeness, we want to mention that we see two directions in which our method can be extended promising a higher speedup in theory, although they are beyond the scope of our manuscript and subject to future work. For this, we note that the cubic scaling of DAMPF is with respect to the dimension of the Hilbert space of the electronic Hamiltonian, i.e. $O((\dim H_{el})^3)$, and our manuscript considered the single-excitation-manifold, where $\dim H_{el} = N$ with N electronic sites.

One possible extension are models with multiple excitations, where $\dim H_{el} = 2^N$ leads to an exponential scaling in classical simulations, while in principle the linear scaling of our algorithm persists. The other extension involves a second qubit register of electronic sites to model electron-hole dynamics, such that for every site both HOMO and LUMO levels are encoded (in the present work, we only regarded electron dynamics in the LUMO levels, since hole dynamics in the HOMO levels are much slower and therefore negligible for the P3HT:fullerene material we considered). A simple model with a single electron-hole pair would require $\dim H_{el} = N^2$, showing a steeper polynomial increase for classical methods, while again in principle the linear scaling of our algorithm persists.

We briefly added these ideas in the Discussion section (Lines 680-682 and 736-741):

“We note that while in general the complexity of DAMPF scales with $(\dim H_{el})^3$, the single-excitation-manifold yields $\dim H_{el} = N$.”

“Beyond the model studied here, extensions of our approach to include arbitrary numbers of excitations or to include electron-hole dynamics with a second register of site qubits promise an even larger speedup over classical methods like DAMPF, that scale with $(\dim H_{el})^3$.”

- 2) **(Rev.1-b)** *The authors mentioned that the bond dimension grows exponentially in the non-perturbative regime for the DAMPF method. (Exponential in time?) Is there a reference for this scaling? My understanding of tensor-network methods is that strong coupling does not necessarily mean they no longer work.*

We thank the reviewer for pointing out that our statement about the growth of the bond dimension χ and the breakdown of classical methods was imprecise. To clarify, we are referring to an exponential growth with respect to simulation time, as we will elaborate below based on [Ulrich Schollwöck, *Annals of Physics*, Volume 326, Issue 1, 2011, 96-192]. Furthermore, we agree that strong vibronic coupling does not necessarily imply a breakdown of tensor-networks methods, especially in perturbative regimes (e.g. when the electronic interaction is much smaller than vibronic coupling $J \ll g_0$). We meant to refer to the non-perturbative regimes characterized by weak damping and vibronic coupling strengths that are comparable to electronic interactions ($J \sim g_0$), which require high bond dimensions for accurate simulations as explained below. We corrected our statement as follows (Lines 724-735):

“Significantly, classical simulations quickly become intractable in non-perturbative regimes where vibronic coupling is comparable to the strength of electronic interactions ($g_0 \sim J$) and oscillators are weakly damped ($\gamma \ll g_0, J$) [39, 59], as exemplified in DAMPF, where the bond dimension that is required to stay below a certain error threshold is determined by the balance between the spread of quantum correlations (entanglement) and local dissipation [60]. In contrast, the quantum algorithm is independent of the bond dimension χ as the quantum processor naturally encodes the full vibronic state in the qubit-mapped Hamiltonian.”

According to the Lieb-Robinson theorem, the entanglement entropy $S(t)$, that characterizes the amount of entanglement between electronic sites and vibronic oscillators, is bound by $S(t) \leq S(0) + ct$, where c captures the propagation speed of entanglement which depends on the strength of electronic interactions in comparison to local dissipation. Since the bond dimension $\chi(t)$ required to represent the quantum state at time t at a fixed numerical accuracy scales as $\chi(t) \propto 2^{S(t)}$, linear growth of $S(t)$ implies exponential growth of χ . We note that the exponential increase especially holds for early times and factorized system-environment initial states with low entropy.

In simulations of open systems, as we consider in our work, the maximum value of χ is determined by a balance of entanglement spreading and dissipation, which acts as a counterforce that steadily reduces entanglement. Dissipation can drastically lower the required bond dimension (see Figure Rev.1-b) and make tensor-network simulations classically feasible. Nonetheless, in the regime of weak damping rates and fast-growing system-bath entanglement (when vibronic coupling and electronic interaction are comparable), bond dimensions χ often remain classically intractable.

[editorial note: third party material redacted]

Figure Rev. 1-b: Growth of the bond dimension in DAMPF over simulation time, with and without dissipation to counteract system-bath entanglement. [Taken from *PRL* 123, 100502 (2019)]

3) **(Rev.1-c)** *It is also important to be aware that the comparison is not apple-to-apple, given that the analog part (ancilla qubit dissipation) is uncontrollable and may not even be completely known. Realistically, the qubit decay rates fluctuate and drift in time, and thus, even a complete calibration (if possible) would not allow the exact decay rate distribution at the time of the experiment to be known. While an upper bound on a relevant characteristic decay time can be estimated, the proposed approach cannot provide a quantitatively accurate result for a specific target set of decay rates. While the authors suggest that limited control is possible by engineering the total time and time step, this will not allow any control of the actual decay rate distribution. In this regard, the complexity analysis is unfair as the proposed protocol and the DAMPF method do not make the same calculation.*

We acknowledge the Reviewer's point regarding the tunability of damping rates. We agree that, unlike DAMPF, which can simulate arbitrary target decay rates γ_n at every site, our quantum approach is constrained to the intrinsic qubit dissipation rates determined by hardware coherence times and the Trotter time step, and that these rates fluctuate and cannot precisely be controlled.

Nevertheless, we argue that the complexity comparison remains meaningful within a certain regime of early simulation times, before inhomogeneities in the qubit dissipation rates have a noticeable impact on the system dynamics. For instance, in the case of equal target damping rates considered in our manuscript, our qubit selection scheme emphasized similar coherence times in oscillator qubits. While residual inhomogeneities in qubit dissipation rates cannot be avoided, they do not impact system dynamics for simulation times $T \lesssim 1/\Gamma_{QC}$, when the precise distribution of hardware damping rates is not yet resolved.

For simulations targeting a specific set of decay rates γ_n , accurate results would in principle be possible if the available dissipation rates on the hardware are smaller than all γ_n , with the remaining dissipation supplied artificially. However, this requires active tuning of individual qubit decay rates, which is beyond the scope of the present work. Platforms offering greater flexibility in engineered dissipation, such as trapped-ion systems, would be better suited for this purpose.

We have added the following clarification in the discussion (Lines 761-775):

“While we above compare our approach to classical methods such as DAMPF, which can target arbitrary decay rates for each oscillator, the damping rates in superconducting processors are not freely tunable: they are determined by the intrinsic qubit dissipation, governed by hardware coherence times T_1 , T_2 and the Trotter time step. For the target model considered here with equal damping rates on each oscillator, our qubit selection strategy addresses this constraint by prioritizing oscillator qubits with similar and maximally long coherence times. The residual

inhomogeneities in dissipation rates remain inconsequential for simulation times $T \lesssim 1/\Gamma_{QC}$, when the precise distribution of hardware damping rates is not yet resolved by the dynamics. “

- 4) **(Rev.1-d)** *In my opinion, it will be very helpful for the authors to make a brief discussion on whether their approach can be readily generalized to simulate 2D models, and to other noise models (maybe shared baths). This discussion will make the approach more toward a toolbox as proposed, rather than a simulation of a specific problem.*

We thank the reviewer for this suggestion, which we believe indeed strengthens the manuscript. In the following we discuss how our approach generalizes to 2D models and to other noise models.

2D models: Our approach extends naturally to 2D lattice geometries. This leads into an interesting direction, where an advantage over classical tensor network methods can become more pronounced: The area law for short-ranged Hamiltonians with a gap dictates that the entanglement entropy is proportional to the surface at the boundary of a bipartition $A|B$ of the system. In two dimensions, the entanglement entropy is no longer constant (in contrast to 1D models), leading to substantially higher bond dimensions. Our quantum simulation approach, by contrast, does not suffer from this dimensional bottleneck: Platforms with more flexible connectivity, such as trapped ions with long-range coupling or neutral atoms with reconfigurable 3D geometries, could implement such models naturally. Furthermore, quantum hardware with 2D-topology with local degrees of freedom (e.g. qudits or native oscillators) are promising for this task. Even on superconducting qubit square-lattice architectures (as considered in Fig. 7), the electronic sites can be arranged in clusters together with their locally coupled oscillators and mapped onto a 2D hardware topology, such that a model of $N \times N$ sites, each locally coupled to M oscillators with N_b levels each, requires $O(MN_b)$ SWAP layers in the Trotter circuit, i.e. the scaling would be independent of the system size N .

Shared baths and correlated noise: Our current model assigns an independent bath oscillator to each electronic site. Shared bath models, where multiple sites couple to a common environment, could be implemented by coupling several electronic sites to the same hardware oscillator. The main requirement is sufficient connectivity between the shared oscillator and the relevant sites, which again favors platforms with flexible coupling topologies. We note that shared baths introduce environment-mediated correlations between sites, which are particularly challenging for classical methods but are captured naturally in our full quantum simulation.

We summarized these points in the discussion section (Lines 741-754):

“Furthermore, our approach can be generalized to 2D lattice geometries, where the entanglement area law leads to even larger bond dimensions that challenge classical tensor network methods, while our quantum simulation does not suffer from this dimensional bottleneck, in particular when employing platforms with flexible coupling topologies: trapped ions with long-range connectivity and neutral atoms with reconfigurable geometries can accommodate 2D layouts. Similarly, shared bath models, where multiple electronic sites couple to a common environment, can be implemented efficiently on such platforms by routing several sites to the same oscillator, capturing environment-mediated correlations that are particularly costly to treat classically.”

Reply to Reviewer #2:

I am satisfied with the authors' response. I think this article serves as a useful demonstration of how to perform quantum simulations of open quantum systems, and will be of great interest to the community.

We thank the reviewer for their time, for their constructive feedback and positive judgement.

Reply to Reviewer #3:

I would like to thank the authors for considering and replying to my comments. For example, I would like to start by further thanking the authors for updating the title which is now, in my opinion, a more precise representation of this work.

5) **(Rev.3-a)** *However, I have to mention that I still have some doubts about the generality of some of the overall claims. For example, even in the very abstract, reference to scalability is still present as*

"our approach enables scalable simulations of non-Markovian dynamics on near-term quantum hardware."

Some of the comments provided in the reply helped me to understand the authors' point of view on this point a bit better. Citing from the reply:

"We emphasize that the primary challenge in simulating open quantum systems lies not in the complexity of the vibrational environment per se, but in the scaling of the system size;"

and

"The goal of our manuscript was to explore increasing system sizes and the feasibility on real quantum hardware."

It seems to me that these sentences well summarize the point of view of the authors, mostly concerned about the scaling in terms of system size other than in the complexity of environmental effects. However, while it could be the case that the system size scaling is the primary challenge for this specific case, I would argue that it is not really true for general non-Markovian open systems.

To be more specific, in the comparison with the DAMPF algorithm, the scaling with respect to the system size is reduced from $O(N^3)$ to $O(N)$. While the scaling in DAMPF could be optimized to $O(N \log(N))$, this does represent an improvement, always justified by the direct use of quantum hardware to represent the closed system, fully in line with the stated authors' goal. Importantly, the authors do also mention an improvement in terms of environmental properties, i.e., the scaling with respect to the pseudomodes truncation N_b from quadratic to linear, since each of the energy levels is mapped to a single qubit.

However, while I can agree with the scalability in terms of the system size, the reliance on a very specific choice for the environment makes it difficult for me to justify the generality of the scaling in terms of environmental properties. In other words, the authors' goal about optimizing the scaling in terms of system size is, in my opinion, more suited to the analysis of quantum algorithms for closed systems rather than open, making it difficult for me to support the point of view behind the claim "our approach enables scalable simulations of non-Markovian dynamics".

Trying to be more specific, the reported estimates for the scaling in terms of environmental properties does rely on a specific model in which all the effects of the environment can be encoded

using a single locally-dissipative harmonic mode for each system site. It is not clear to me that this scaling would still be valid for more general non-Markovian dynamics such as (i) deeper in non-perturbative regimes where the qubits non-linearity could hinder the validity of the effective model or (ii) in a regime relying on non-local coupling in the system-sites labeling (which might even hinder the scaling with respect to the system size) or (iii) for broader local environments (where $C(t)$ is not just the Fourier transform of $J(\omega)$).

While I understand that all of the above is beyond the scope of this work, my opinion is only intended to support my personal opinion that general claim about scalability might, possibly, be better to be toned down a little or postponed until fully supported by future analysis.

In summary, while I do appreciate the framework provided for the potential simulation of non-Markovian open quantum systems and the overall high quality of this work, I also do think that these results rely on a very specific choice of open quantum system. As a consequence, I would argue that the generality of the overall claims might also be better to be adjusted accordingly.

We thank the reviewer for their in-depth assessment and agree with them that the experimental demonstrations presented here do not yet constitute empirical proof of scalability across general non-Markovian regimes. Therefore, we have modified the abstract to reflect the scope of the demonstrated results rather than the theoretical outlook. Specifically, we now emphasize the reproduction of non-Markovian effects and deliberately avoid any claims about favorable scalability:

"We validate our strategy by resolving the vibronic transfer spectra of a one-dimensional donor-acceptor chain on IBM superconducting processors, reproducing non-Markovian dynamics and scaling the chain length up to 10 electronic sites, an unprecedented scale for chemical dynamics on quantum computers."

Additionally, we agree that the algorithm is tailored to a particular family of open quantum systems (those governed by Hamiltonians with linear-vibronic coupling), and it does not represent a general toolbox for arbitrary system-environment coupling structures. We have modified the following sentence in the abstract to clarify this point:

"We present a framework for the digital-analog simulation of open quantum systems governed by Hamiltonians with linear-vibronic coupling (LVC) and structured vibrational environments."

Furthermore, we checked that the manuscript contains no further claims regarding general non-Markovian dynamics. To complete this reply, we want to comment on the reviewer's points about the degree of non-Markovianity accessible to our method, as we consider them interesting challenges. While these three points require their own analysis in future work, we see a chance to extend our method and therefore address them individually in the following:

- i. "Non-perturbative regimes" involving higher-order terms beyond linear vibronic coupling to model anharmonic oscillators: Such models are unfortunately outside the scope of the pseudomode formalism, which rely on the assumption of Gaussian environments.
- ii. "Non-local coupling in system sites": Long-range interactions beyond nearest-neighbor hopping can be included by introducing SWAP gates in the Trotter circuit to bring non-neighboring sites into adjacency for two-qubit gates. Interactions of up to m -th order neighbors can be included at the expense of $O(m^2)$ layers of SWAP gates. For the presented case $m=1$, a single layer was sufficient. Furthermore, for non-local site-bath interactions, we refer to our reply to Rev.1-d on shared baths.

- iii. “Broader local environments”: We note that within the harmonic bath framework at finite temperatures, $C(t)$ differs from a (one-sided) Fourier transform of $J(\omega)$ precisely because it additionally encodes the thermal occupation of bath modes. Our framework can account for this by fitting the model's spectral density to the target correlation function at a given temperature. Capturing environments with richer spectral structure (e.g., multiple peaks in $J(\omega)$) requires scaling to multiple oscillators per site, as depicted in Figure 7c in the manuscript.

6) **(Rev.3-b)** *As a minor additional comment, I was wondering about the reply to referee 2 question 13, also related to my point (iii) above. As far as I know, $C(t)$ is not the Fourier transform of the physical spectral density $J(\omega)$ at zero temperature. I would then thereby want to ask the authors whether they could specify the precise meaning of the term "Fourier pair" used in the manuscript.*

The reviewer is right in pointing out that at zero temperature, $C(t)$ is the *one-sided* Fourier transform of $J(\omega)$. We acknowledge that the usage of the term Fourier pair was imprecise.

We changed the manuscript accordingly (Lines 808-811):

“Here we fix the temperature to zero, where the bath correlation function $C(t)$ is the one-sided Fourier transform of the spectral density $J(\omega)$. “